# Retinal stem cells modulate proliferative parameters to coordinate post-embryonic morphogenesis in the eye of fish

**Erika Tsingos[1]\*, Burkhard Höckendorf[1†], Thomas Sütterlin[2], Stephan Kirchmaier[1‡], Niels Grabe[2], Lazaro Centanin[1], Joachim Wittbrodt[1]\***

[1]Centre for Organismal Studies Heidelberg, Heidelberg University, Heidelberg, Germany; [2]National Center for Tumor Diseases, Hamamatsu TIGA Center, Bioquant, Heidelberg University, Heidelberg, Germany

**Abstract** Combining clonal analysis with a computational agent based model, we investigate how tissue-specific stem cells for neural retina (NR) and retinal pigmented epithelium (RPE) of the teleost medaka (*Oryzias latipes*) coordinate their growth rates. NR cell division timing is less variable, consistent with an upstream role as growth inducer. RPE cells divide with greater variability, consistent with a downstream role responding to inductive signals. Strikingly, the arrangement of the retinal ciliary marginal zone niche results in a spatially biased random lineage loss, where stem- and progenitor cell domains emerge spontaneously. Further, our data indicate that NR cells orient division axes to regulate organ shape and retinal topology. We highlight an unappreciated mechanism for growth coordination, where one tissue integrates cues to synchronize growth of nearby tissues. This strategy may enable evolution to modulate cell proliferation parameters in one tissue to adapt whole-organ morphogenesis in a complex vertebrate organ.
DOI: https://doi.org/10.7554/eLife.42646.001

**\*For correspondence:**
Erika.tsingos@cos.uni-heidelberg.
de (ET);
jochen.wittbrodt@cos.uni-
heidelberg.de (JW)

**Present address:** [†]Janelia
Research Campus, Howard
Hughes Medical Institute,
Ashburn, United States;
[‡]Deutsches
Krebsforschungszentrum,
Heidelberg, Germany

**Competing interests:** The
authors declare that no
competing interests exist.

**Reviewing editor:** Raymond E
Goldstein, University of
Cambridge, United Kingdom

## Introduction

To maintain proper proportions, growth must be regulated at the level of the whole body, the size of each organ, and the size of tissues within an organ (*Roselló-Díez and Joyner, 2015*). Some regulatory mechanisms are shared, while others are specific to each level or to particular organs (*Lui and Baron, 2011*; *Roselló-Díez and Joyner, 2015*). Systemic signals couple nutrition to growth to coordinate growth of all organs at the organismal level (*Buchmann et al., 2014*; *Droujinine and Perrimon, 2016*). In addition to extrinsic systemic factors, transplantation experiments showed that many organs, including the eye, grow autonomously according to intrinsic factors (*Wallman and Winawer, 2004*; *Roselló-Díez and Joyner, 2015*). Growth coordination mechanisms have been studied at the level of the whole organism and inter-organ communication (*Buchmann et al., 2014*; *Droujinine and Perrimon, 2016*), but feedback mechanisms between constituent tissues of an organ remain largely unexplored both experimentally and at a conceptual level (*Buchmann et al., 2014*).

Teleost fish grow throughout their lives, increasing massively in size (*Johns and Easter, 1977*). The teleost medaka (*Oryzias latipes*) grows roughly ten-fold from hatching to sexual maturity within 2–3 months (*Figure 1—figure supplement 1A*). Unlike embryonic morphogenesis, during post-embryonic growth all organs must scale with the increasing body size while fully functioning. In the eye, continuous growth must be additionally balanced with continuous shape-keeping: Proper optics, and thus vision, requires a precise 3D shape. Highly visual shallow water fish such as medaka have near-perfect hemispherical eyes (*Fernald, 1990*; *Nishiwaki et al., 1997*; *Beck et al., 2004*). The growth rates of all eye tissues must perfectly match, otherwise the organ would deform, akin to

**eLife digest** By the time babies reach adulthood, they have grown many times larger than they were at birth. This development is driven by an increase in the number and size of cells in the body. In particular, special types of cells, called stem cells, act as a reservoir for tissues: they divide to create new cells that will mature into various specialized structures.

The retina is the light-sensitive part of the eye. It consists of the neural retina, a tissue that contains light-detecting cells, which is supported by the retinal pigment epithelium or RPE. In fish, the RPE and neural retina are replenished by distinct groups of stem cells that do not mix, despite the tissues being close together.

Unlike humans, fish grow throughout adulthood, and their eyes must then keep pace with the body. This means that the different tissues in the retina must somehow coordinate to expand at the same rate: otherwise, the retina would get wrinkled and not work properly. Tsingos et al. therefore wanted to determine how stem cells in the neural retina and RPE co-operated to produce the right number of new cells at the right time.

First, stem cells in the eyes of newly hatched fish were labelled with a visible marker so that their divisions could be tracked over time to build cell family trees. This showed that stem cells behaved differently in the neural retina and the RPE. Computer simulations of the growing retina explained this behavior: stem cells in the neural retina were telling the RPE stem cells when it was time to divide. Combining results from the simulations with data from the experiments revealed that a stem cell decided to keep up dividing partly because of its position in the tissue, and partly because of random chance.

To be healthy, the body needs to fine-tune the number of cells it produces: creating too few cells may make it difficult to heal after injury, but making too many could lead to diseases such as cancer. Understanding how tissues normally agree to grow together could therefore open new avenues of treatment for these conditions.

DOI: https://doi.org/10.7554/eLife.42646.002

a bimetallic strip. Thus, the eye of fish provides an excellent system to explore how anatomically and functionally distinct tissues coordinate to grow and maintain the shape of an organ in functional homeostasis (*Johns and Easter, 1977*; *Centanin et al., 2014*).

The vertebrate eye consists of multiple concentric tissues, including the neural retina (NR) and the retinal pigmented epithelium (RPE) (*Figure 1A*; *Table 1*). In fish and amphibians, these tissues grow from a ring-shaped stem cell niche in the retinal periphery: the ciliary marginal zone (CMZ) (*Johns, 1977*; *Harris and Perron, 1998*; *Amato et al., 2004*). The CMZ can be subdivided into a peripheral stem- and a central progenitor cell domain; stem cells are believed to have the potential for indefinitely many cell divisions while progenitor cells divide only a handful of times (*Raymond et al., 2006*; *Centanin et al., 2014*; *Wan et al., 2016*; *Shi et al., 2017*). At the very periphery of the CMZ, about 5 rows of cells express the stem cell marker retina-specific homeobox gene 2 (Rx2) (*Reinhardt et al., 2015*; *Wan et al., 2016*; *Tang et al., 2017*). The CMZ is a bi-partite niche, with tissue-specific stem cells for NR and RPE (*Shi et al., 2017*). In medaka, stem cells for NR and RPE are strictly separate, as demonstrated by transplantations at blastula stage and genetic recombination after hatching (*Centanin et al., 2011*; *Centanin et al., 2014*). Thus, medaka NR and RPE are independently growing tissues with identical topology.

As a population, CMZ cells appositionally add new cells in concentric rings as shown by label incorporation with thymidine analogues (*Johns, 1977*; *Centanin et al., 2011*). Individual stem cells labelled by genetic markers form clonal progeny in so-called Arched Continuous Stripes (ArCoS; *Figure 1B*) (*Centanin et al., 2011*; *Centanin et al., 2014*). Medaka NR stem cells produce the full complement of neuronal cells in apico-basal clonal columns (*Figure 1—figure supplement 2A'–B*) (*Centanin et al., 2011*; *Centanin et al., 2014*; *Lust and Wittbrodt, 2018*). These differentiated retinal cells grow little in size (*Johns, 1977*), retain their relative position over time (*Johns, 1977*; *Centanin et al., 2011*), and have negligible death rates (*Johns and Easter, 1977*; *Stenkamp, 2007*). Thus, the only parameter available to NR and RPE to coordinate their growth rates is the proliferation of the tissue-specific CMZ stem cells.

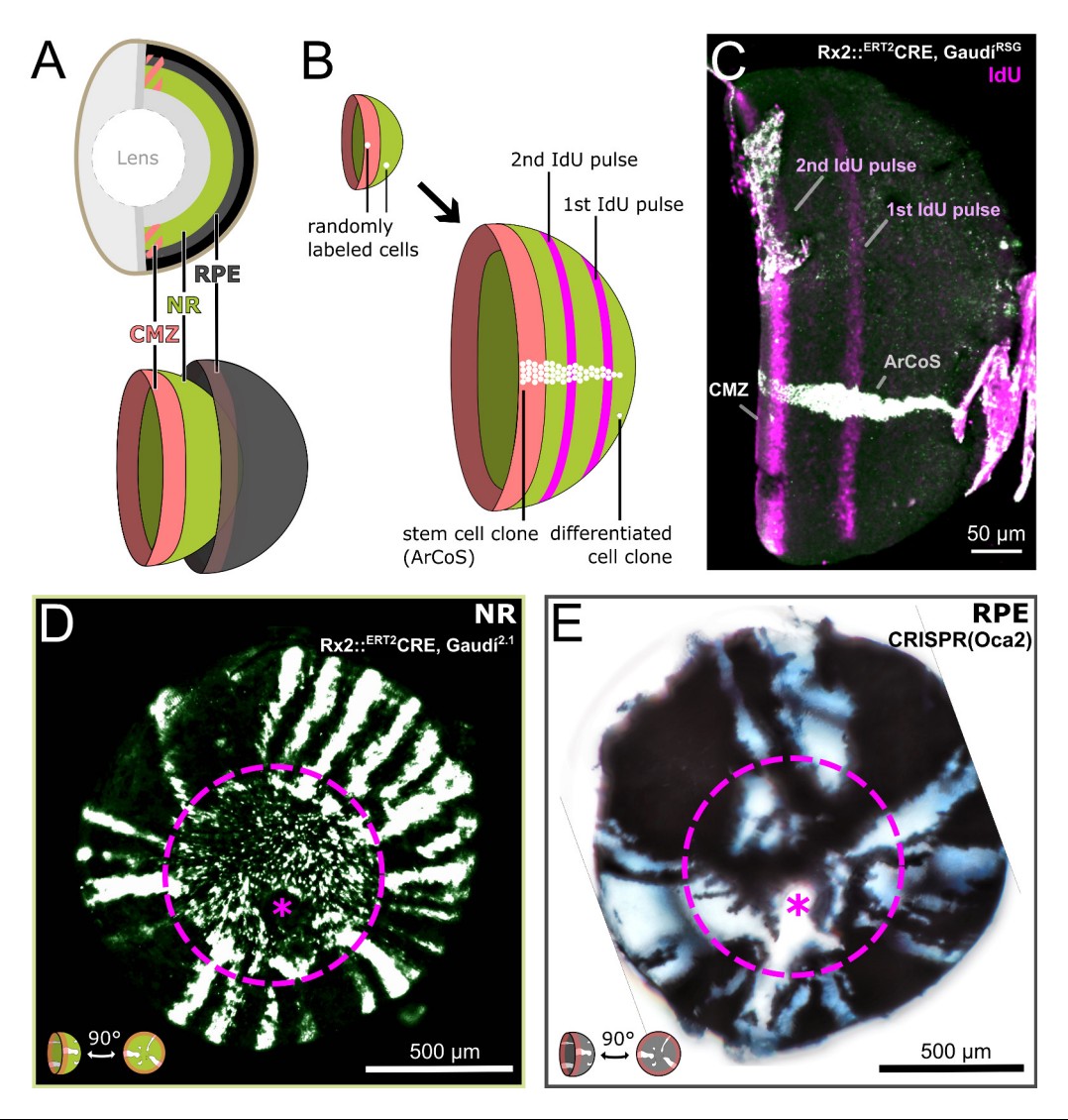

**Figure 1.** Clonal labelling enables analysis of growth patterns in NR and RPE. (**A**) Schematic anatomy of the fish eye. (**B**) Growth patterns of retinal cell population (concentric rings) and individual clones. (**C**) False color immunostained NR of 3 week old Rx2::ERT2Cre, Gaudi^RSG fish with ArCoS and concentric rings of IdU-labelled cells. Overnight IdU pulses were at 1 and 2.5 weeks of age. Leftover undissected autofluorescent tissue fragments cover the far right of the cup-shaped retina. (**D**) Proximal view of clones induced in the NR of Rx2::ERT2Cre, Gaudi^2.1 fish. Maximum projection of confocal stack of GFP immunostaining in false colors. (**E**) Proximal view of unpigmented lineages induced in the RPE by mosaic bi-allelic knockout of Oca2 using CRISPR/Cas9. Focused projection of brightfield focal stack. Images in (**D**) and (**E**) have been rotated to place the optic nerve exit (pink asterisk) ventrally; the embryonic retina is circled with a pink dashed line.

DOI: https://doi.org/10.7554/eLife.42646.004

The following figure supplements are available for figure 1:

**Figure supplement 1.** The retinal radius represents a temporal axis.
DOI: https://doi.org/10.7554/eLife.42646.005

**Figure supplement 2.** NR ArCoS form narrow columns spanning all NR layers.
DOI: https://doi.org/10.7554/eLife.42646.006

**Table 1.** List of abbreviations used throughout the main text.

| NR | neural retina |
|---|---|
| RPE | retinal pigment epithelium |
| CMZ | ciliary marginal zone |
| Rx2 | retina-specific homeobox gene 2 |
| ArCoS | arched continuous stripes |
| GFP | green fluorescent protein |
| Oca2 | oculo-cutaneous albinism 2 |
| IdU | 5-Iodo-2′-deoxyuridine |

DOI: https://doi.org/10.7554/eLife.42646.003

Stem cells have long been defined by an unlimited self-renewal capacity (*Watt and Hogan, 2000*; *Clevers and Watt, 2018*). Two general strategies underlie long-term maintenance of stem cells: 1) a deterministic model where every single division produces a stem- and a progenitor daughter cell ('invariant asymmetry'); and 2) a stochastic model where cells divide symmetrically, and the daughter cells have a probability to stay as stem cells or commit to a progenitor fate ('neutral drift') (*Watt and Hogan, 2000*; *Clevers and Watt, 2018*). One tenet of this model is neutral competition: Stem cells randomly displace one another, resulting in the 'loss' of lineages where all progeny commit to a progenitor fate until the entire niche is occupied by a single clone (*Colom and Jones, 2016*; *Clevers and Watt, 2018*).

Strikingly, the medaka retina diverges from the neutral drift model. The CMZ maintains a polyclonal stem cell population for both the NR and the RPE, and in particular NR stem cells undergo asymmetric self-renewing divisions throughout the life of the animal (*Centanin et al., 2011*; *Centanin et al., 2014*). It remains unclear whether stem cell proliferation in the CMZ follows a purely deterministic model, or whether it follows a strategy in-between invariant asymmetry and neutral drift.

In this work we combine *in vivo* and *in silico* clonal analysis in the NR and RPE of medaka to address how these tissues coordinate their growth rates. We find that RPE stem cells have highly variable cell division timing consistent with a downstream role in the control hierarchy, whereas NR stem cells display less variability consistent with an upstream role in inducing growth in nearby tissues. Our simulation predicts that the spatial segregation of stem and progenitor CMZ domains is an emergent property, as the topology of the retinal niche preconditions the retina to a spatially biased neutral drift. NR stem cells deviate from a purely random drift model by preferential division axis orientation and differential modulation of division parameters along the CMZ circumference. We propose that during post-embryonic growth of the teleost eye, the NR CMZ forms a hub for integrating external and internal stimuli that affect cell division parameters, which ultimately direct the growth and shape of the entire eye.

## Results

### Clonal analysis indicates NR and RPE follow different post-embryonic growth modes

Retinal cells follow an exquisite spatiotemporal order (*Figure 1B–C*, *Figure 1—figure supplement 1B*). Thus, clones derived from stem cells are a frozen record of past cell divisions (*Centanin et al., 2011*; *Centanin et al., 2014*), offering a window of opportunity to study stem cell properties in the NR and RPE.

We experimentally generated NR ArCoS by randomly labelling individual NR stem cells using the Rx2::[ERT2]Cre, Gaudí[2.1] line in hatchling medaka, and analyzing the eyes in adult fish as previously described (*Centanin et al., 2014*; *Reinhardt et al., 2015*). The Rx2 promoter drives the inducible Cre recombinase in stem cells at the very periphery of the CMZ (*Reinhardt et al., 2015*). A recombined stem cell generates a stripe of GFP-positive progeny in an otherwise GFP-negative retina (*Centanin et al., 2014*). In proximal view, NR ArCoS emanated as rays from the central embryonic

retina, the part of the eye that was already differentiated at the timepoint of Cre-mediated recombination (*Figure 1D*).

We visualized RPE ArCoS by mosaic knockout of pigmentation using CRISPR/Cas9 targeted to the gene oculo-cutaneous albinism 2 (Oca2), which is required for melanosome maturation (*Fukamachi et al., 2004*; *Lischik et al., 2019*). RPE stem cells with a bi-allelic mutation in Oca2 generate unpigmented stripes, analogous to RPE ArCoS obtained by transplantation (*Centanin et al., 2011*). RPE ArCoS frequently branched, forming irregular stripes variable in size and shape (*Figure 1E*). These qualitative differences in clonal pattern suggested that despite their identical topology, the division behavior of NR and RPE stem cells differed.

## A 3D agent based model of retinal tissues

Clonal data generates a distribution of outcomes that is challenging to analyse and easy to misinterpret (*Klein et al., 2007*). The curved retinal surface and spatial extent of the niche pose a further challenge. We overcome these challenges by comparing experimental clonal data with simulated clonal data from a 3D agent based cell-center overlapping spheres model built in the platform EPISIM (*Sütterlin et al., 2013*; *Sütterlin et al., 2017*; *Sütterlin, 2019*). This modelling technique represents cells as discrete objects (*e.g.* spheres) that physically interact through forces acting on the cell centers; the spheres are allowed to slightly overlap to simulate cell deformability and allow a tight cell packing (*Sütterlin et al., 2013*; *Sütterlin et al., 2017*). This level of abstraction is ideally suited to the tightly packed pseudocrystalline mosaic of retinal cells (*Johns, 1981*; *Nishiwaki et al., 1997*; *Pérez Saturnino et al., 2018*), and has been used previously to model clonal data in skin and gut epithelia (*Osborne et al., 2010*; *Buske et al., 2011*; *Li et al., 2013*).

Our retinal tissue model consists of a layer of spheres (representing either NR or RPE cells) on a hemisphere (representing the rest of the organ that is not explicitly modelled; *Figure 2A*). The RPE is a monolayer, thus each model cell corresponds to one RPE cell. In the NR, CMZ stem cells form a monolayer, and their differentiated progeny arrange in multiple neuronal layers (*Johns, 1977*; *Raymond et al., 2006*). We observed that clonal progeny of CMZ stem cells retained close proximity with little spread tangential to the retinal surface, forming clonally related 'columns' (*Figure 1—figure supplement 2A'-B*) (*Centanin et al., 2011*; *Centanin et al., 2014*; *Lust and Wittbrodt, 2018*). We took advantage of this fact to abstract each differentiated clonal column as a single cell in the simulation.

*In vivo*, the spatial extent of the CMZ stem cell domain is believed to be defined by cues such as nearby blood vessels (*Wan et al., 2016*; *Tang et al., 2017*). Therefore, we defined the virtual stem cell domain with a fixed size of 25 µm, that is 5 rows of cells, reflecting the endogenous scale of the Rx2-expressing CMZ domain (*Reinhardt et al., 2015*; *Wan et al., 2016*; *Tang et al., 2017*). *In vivo*, NR stem cells divide predominantly asymmetrically, but also undergo symmetric divisions (*Centanin et al., 2014*). The rates of asymmetric and symmetric divisions are unknown; likewise, it is unknown whether these rates are deterministically defined or an emergent property of an underlying stochastic system. Since stochastic cell divisions successfully describe the proliferation of committed retinal progenitor cells in larval zebrafish (*Wan et al., 2016*), we used a simple stochastic mechanism for our initial model. Virtual stem cells commit to divide with a fixed probability $p_{\text{division}} = \frac{1}{26}\,\text{h}^{-1}$ and intervals between subsequent cell divisions must fulfill a minimum cell cycle length $t_{\text{cellCycle}} = 24\text{h}$. These values lie within a biologically plausible range estimated from experimentally measured growth rates and a parameter scan of the simulation (Appendix 1 section 3.3). All divisions are symmetric, resulting in two stem cells; cells differentiate and stop cycling when they exit the virtual CMZ after being pushed out by cellular crowding.

To prevent physically implausible cell crowding, cell-center based models include a density-dependent inhibition of cell division (*Pathmanathan et al., 2009*; *Osborne et al., 2017*; *Sütterlin et al., 2017*). In our model, inhibition occurs in cells whose average overlap with all neighbors exceeds a fraction of the cell's diameter given by the model parameter $\delta_{\text{ol\_threshold}}$ (*Figure 2—figure supplement 1*; Appendix 1, section 2.4). Based on *in vivo* observations (*Lyall, 1957*; *Johns, 1977*; *Ohki and Aoki, 1985*), the growing virtual eye gradually moves cells apart as it expands, thus decreasing cell density (*Figure 2—figure supplement 2*; Appendix 1 section 2.2). Continuous proliferation in the CMZ counteracts this decrease *in vivo* (*Johns, 1977*; *Johns and Easter, 1977*); likewise, the ever-increasing virtual cell population optimally fills the hemisphere at all

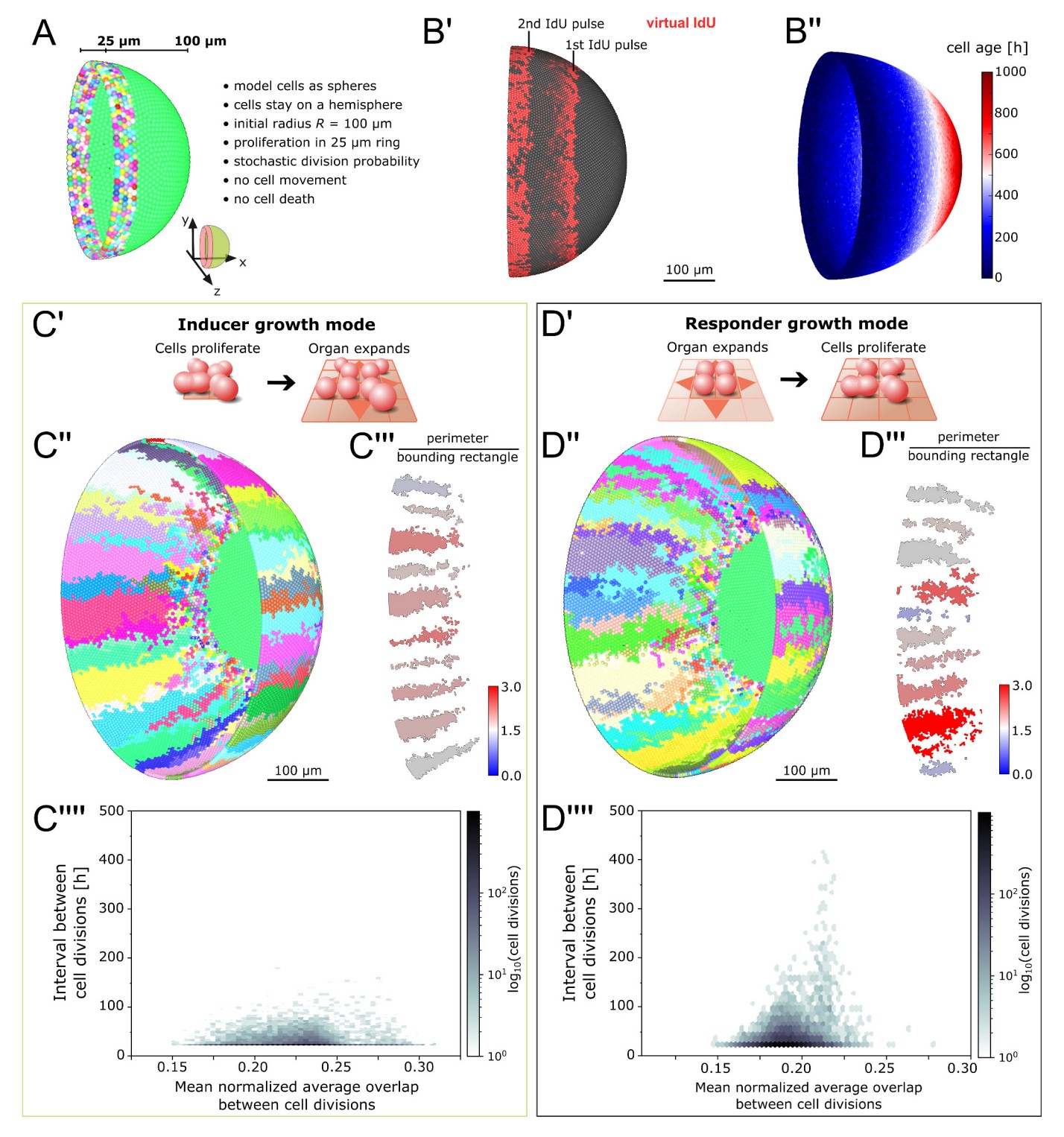

**Figure 2.** Feedback between proliferation and organ growth affects the simulated clonal pattern. (**A**) Initial condition and properties of the agent based model of the growing fish retina. Virtual embryonic retina in light green. CMZ cells are assigned unique colors for virtual clonal analysis. (**B'**) Simulated IdU pulse-chase experiment. First pulse: 200–220 hr, second pulse: 400–420 hr. Screenshot from 435 hr. Virtual cells incorporate IdU when they divide and half of the signal is passed on to each daughter cell. (**B''**) Cell age (hours elapsed since last cell division) forms a gradient with the oldest cells in the virtual embryonic retina. (**C'**) In the inducer growth mode, the modelled tissue signals upstream to drive growth of other tissues in the organ. (**C''**) Representative screenshot of inducer growth mode. (**C'''**) Sample of 10 clones from (**C''**). Colors: ratio of full perimeter by bounding rectangle

*Figure 2 continued on next page*

*Figure 2 continued*

perimeter, a metric for shape complexity. (C'''') Cell division intervals plotted against the mean average overlap. (D') In the responder growth mode, control of the growth of the modelled tissue is downstream of an external signal. (D'') Representative screenshot of the responder growth mode. (D''') Sample of 10 clones from (D'') evaluated by the same shape metric as in (C'''). (D'''') Cell division intervals plotted against the mean average overlap. Note the higher range of values for cells over the threshold overlap of 0.2.

DOI: https://doi.org/10.7554/eLife.42646.007

The following figure supplements are available for figure 2:

**Figure supplement 1.** Process diagram summarizing model decision tree.

DOI: https://doi.org/10.7554/eLife.42646.008

**Figure supplement 2.** Cell position update with radial growth of the simulated retina.

DOI: https://doi.org/10.7554/eLife.42646.009

**Figure supplement 3.** Obstacle cells create an impassable boundary at the hemisphere's edge.

DOI: https://doi.org/10.7554/eLife.42646.010

times (*Video 1*; *Video 2*). Our model distills the complexity of the system and replicates the exquisite spatiotemporal growth order observed *in vivo* (*Figure 2B', B''*).

## Fundamental feedback modes of organ and cell growth impact on clonal patterns

Conceptually, we reasoned that feedback between tissues in an organ can be wired in two fundamental ways: Either the tissue of interest acts upstream to induce growth of other tissues (*Figure 2C'*; 'inducer growth mode'), or, vice versa, the tissue of interest lies downstream of growth cues from another tissue in the organ (*Figure 2D'*; 'responder growth mode'). Possible biological mechanisms for these growth modes could be mechanical, biochemical, or a

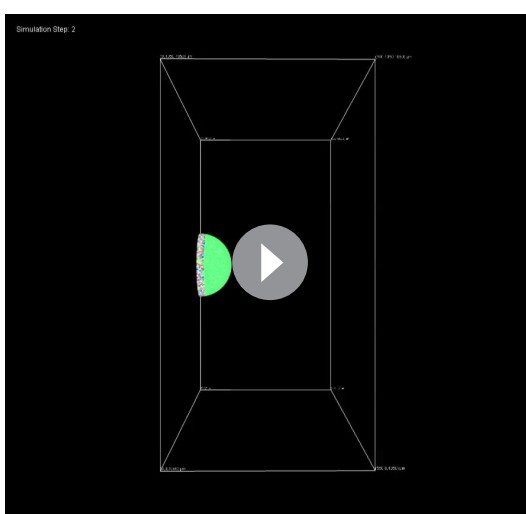

**Video 1.** The simulated retina is always densely covered by cells. Simulation of the responder growth mode illustrating clonal lineage formation while the virtual eye grows. When cells divide they briefly flash white.

DOI: https://doi.org/10.7554/eLife.42646.011

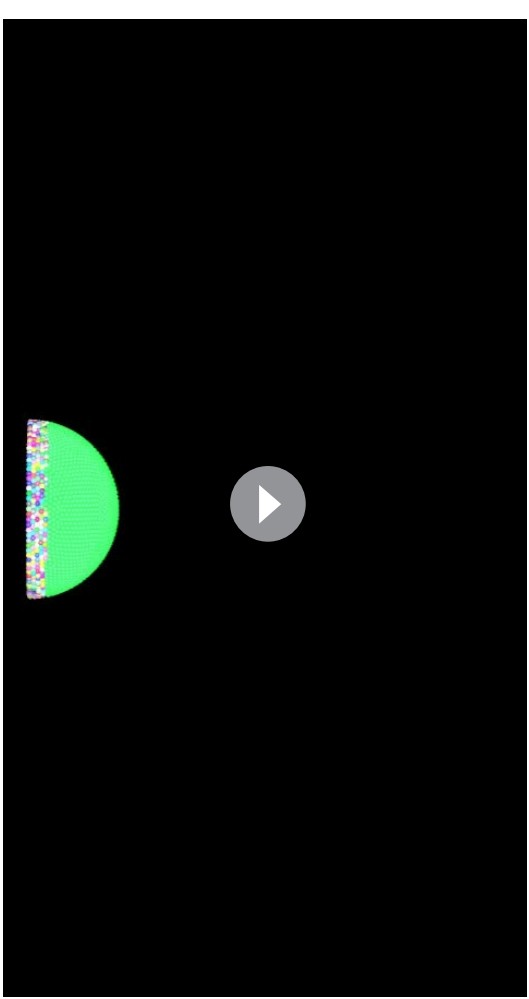

**Video 2.** Lateral view of a simulation of the inducer growth mode. Simulation of the inducer growth mode illustrating clonal lineage formation while the virtual eye grows. When cells divide they briefly flash white.

DOI: https://doi.org/10.7554/eLife.42646.012

combination of both. For example, in the inducer growth mode cells could instruct organ growth by modifying the extracellular matrix or by paracrine signalling (*Buchmann et al., 2014*; *Droujinine and Perrimon, 2016*). These stimuli instruct tissues with the responder growth mode to grow, for example by alleviating contact inhibition or by providing permissive proliferation signals (*Buchmann et al., 2014*; *Droujinine and Perrimon, 2016*). In an organ composed of multiple tissues, one tissue may be the driver for growth, while the rest follows.

We examined how these two conceptual growth modes affected stem cell dynamics in the simulation. In our implementation of the inducer growth mode, an increase in cell number induces growth of the virtual eye's radius (*Appendix 1—equation (5)*). Implicit in this growth mode is the assumption that cell division is not inhibited by the degree of cell crowding normally present in the tissue (otherwise the organ would never grow). Therefore, we set the tolerated overlap threshold $\delta_{ol\_threshold} = 0.4$, a value which we determined by parameter scan to minimize cell division inhibition while preventing physically implausible crowding (Appendix 1, section 3.2).

In the responder growth mode, we let the radius grow linearly over time (*Appendix 1—equation (6)*). In this growth mode, cells must stop dividing until they receive an external stimulus. We take advantage of the pre-existing local density sensing to implement a physical stimulus akin to contact inhibition. Thus, we set the tolerated overlap threshold $\delta_{ol\_threshold} = 0.2$ to maximize cell division inhibition at homeostatic density (Appendix 1, section 3.2). As growth of the hemisphere decreases cell density, cells dynamically respond to growth of the eye by resuming divisions.

In short, the growth modes in our simulation differ only in: 1) the growth equation for the radius of the hemisphere, 2) the value of the threshold parameter $\delta_{ol\_threshold}$ where local cell density inhibits cell divisions (for details, the reader is referred to Appendix 1, sections 2.3; 2.4; and 3.2).

We obtained virtual ArCoS regardless of growth mode (*Figure 2C''', D''*). The growth mode strongly impacted on the shape of ArCoS. Clones in the inducer growth mode formed well-confined stripes with low variation in shape (*Figure 2C'''*). In the responder growth mode, the virtual clones frequently intermingled and broke up into smaller clusters (*Figure 2D'''*). Specifically, the growth modes impacted on variation in cell division timing (*Figure 2C'''', D''''*). In the responder growth mode, local competition for space increased cell division intervals, particularly among cells exceeding the tolerated overlap threshold $\delta_{ol\_threshold} = 0.2$ (*Figure 2D''''*). Thus, the model predicted distinct levels of variation in cell division timing in retinal tissues following the inducer or responder growth modes.

## NR stem cells have less variable cell division timing compared to the RPE

Since the position of cells in the retina reflects their birth order (*Centanin et al., 2011*; *Centanin et al., 2014*), we reasoned that in the extreme case of no variation in cell division timing, each clone forms a continuous, unbranching stripe (*Figure 3B*, left). In the opposite highly variable case, clones frequently branch or merge into polyclones, as well as fragment into several small patches (*Figure 3B*, right). Thus, with increasing variation in cell division timing, we expect an increasing variation in clone width, and an increasing incidence of clone branching and fragmentation.

To quantitatively underpin our previous observations, we compared simulated clones of the inducer and responder growth modes to clones in the NR and RPE (*Figure 3A', A''*). We circumvented biases associated with fusion and fragmentation of clones by analyzing 'patches', that is contiguous domains of segmented pixels. A patch may entail a (sub-)clone, or multiple clones (*i.e.* a polyclone) (*Figure 3—figure supplement 1*; *Video 3*). To assay our experimental and simulated data, we unrolled the retina with a coordinate transform (*Figure 3—figure supplement 2C*) and quantified three different metrics: patch width variance, branching, and fragmentation.

To assay patch width variance, we aligned and superimposed all patches (*Figure 3C', C''*), and quantified the distribution of maximum patch width (*Figure 3—figure supplement 2A*; *Figure 3—figure supplement 2—source data 3.*). Confirming our previous qualitative observations, NR patches formed a narrow stripe, while the width of RPE patches showed much greater variation (*Figure 3C'*; *Figure 3—figure supplement 2A*). The variance of NR and RPE patches was significantly different at the 0.05 level (p=$3.50 \cdot 10^{-12}$, F-test of equality of variance). In striking agreement to the experimental data, simulated patches in the inducer growth mode had low variation in width, while patches in the responder growth mode spread widely (*Figure 3C''*; *Figure 3—figure*

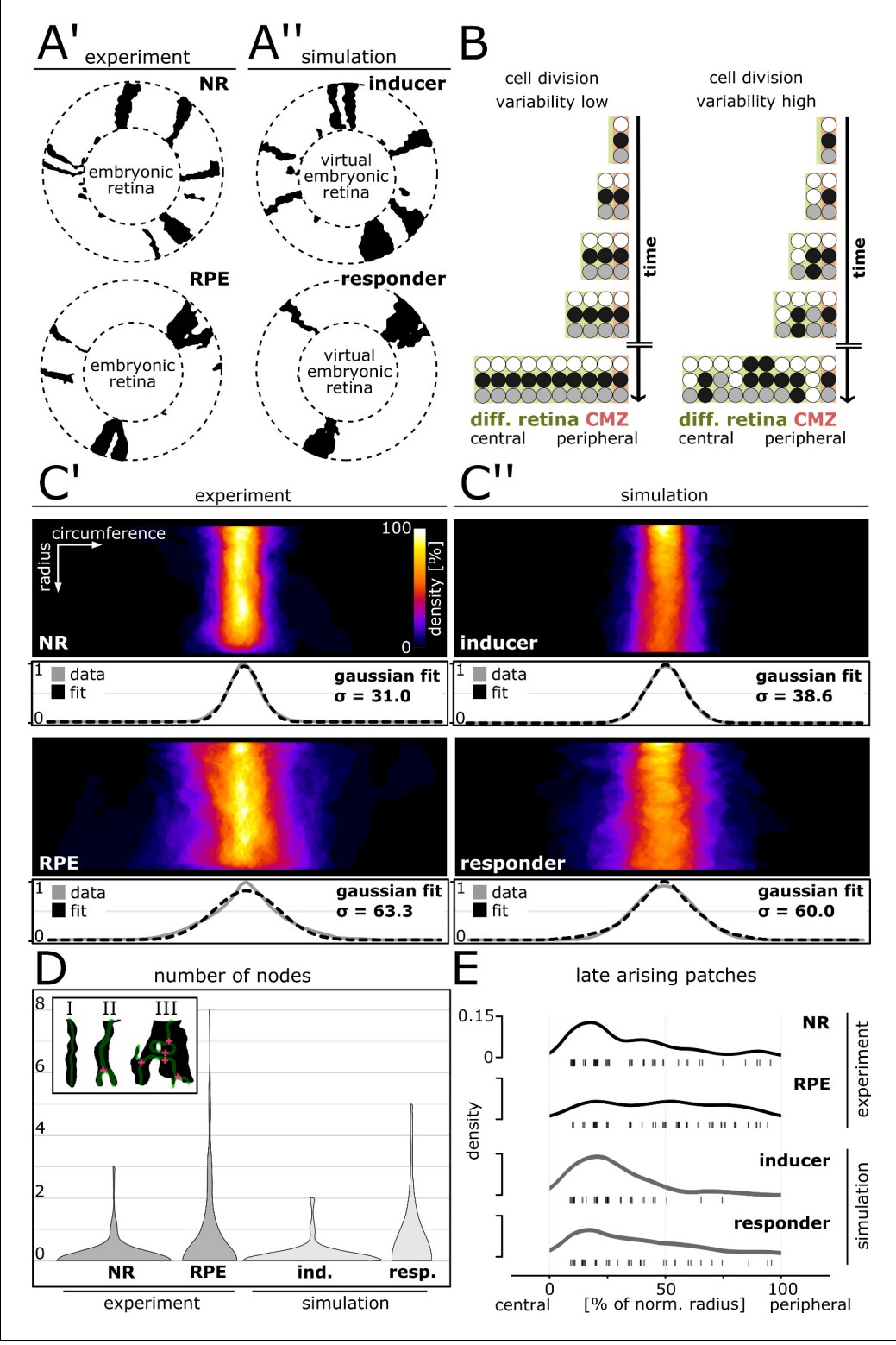

**Figure 3.** Cell division variability is lower in NR and inducer growth mode, higher in RPE and responder growth mode. (A'–A'') Proximal view of segmented patches in adult NR and RPE and simulated patches in inducer and responder growth mode. The central (virtual) embryonic retina was excluded from analysis. (B) Different degrees of variability in cell division timing affect the clone pattern. (C'–C'') Upper panels: Superposition of labelled patches in the NR (n = 156 patches from seven retinae), RPE (n = 142 patches from 10 retinae), inducer growth mode (n = 145 patches from five simulations), and responder growth mode (n = 107 patches from five simulations). The
*Figure 3 continued on next page*

*Figure 3 continued*

radius was normalized to the same length in all samples. Lower panels: Gaussian fits of normalized pixel intensity profiles projected along the vertical axis. σ - Standard deviation of fit. (D) Distribution of number of nodes of skeletonized patches. Inset: Examples of patches without nodes (I), with only one node (II), or with multiple nodes (III). (E) Rug plot showing number of patches that are not connected to the embryonic retina ('late arising patches') at the respective positions along the normalized radius. NR (n = 54 late patches) and inducer growth mode (n = 35 late patches) display a marked peak in the central portion, while RPE (n = 56 late patches) and responder growth mode (n = 37 late patches) have a more uniform distribution.

DOI: https://doi.org/10.7554/eLife.42646.013

The following source data and figure supplements are available for figure 3:

**Source data 1.** Patch outlines.
DOI: https://doi.org/10.7554/eLife.42646.016
**Source data 2.** Patch superposition.
DOI: https://doi.org/10.7554/eLife.42646.017
**Source data 3.** Nodes per patch *Figure 3D*) Counts of number of nodes in each patch for each condition.
DOI: https://doi.org/10.7554/eLife.42646.018
**Source data 4.** Late arising patches *Figure 3E*) Counts of number of patches along normalized radial bins.
DOI: https://doi.org/10.7554/eLife.42646.019
**Source data 5.** Comparison of distribution of number of nodes.
DOI: https://doi.org/10.7554/eLife.42646.023
**Source data 6.** Comparison of distribution of late arising patches.
DOI: https://doi.org/10.7554/eLife.42646.024
**Figure supplement 1.** Relationship between clones, patches, and polyclones.
DOI: https://doi.org/10.7554/eLife.42646.014
**Figure supplement 2.** Distributions of patch width and length in experiment and simulation.
DOI: https://doi.org/10.7554/eLife.42646.015
**Figure supplement 2—source data 1.** Patch width distribution.
DOI: https://doi.org/10.7554/eLife.42646.020
**Figure supplement 2—source data 2.** Patch height distribution.
DOI: https://doi.org/10.7554/eLife.42646.021
**Figure supplement 2—source data 3.** Comparison of variances of maximum patch width distribution.
DOI: https://doi.org/10.7554/eLife.42646.022

*supplement 2A*). The variances in the simulated conditions were significantly different at the 0.05 level ($p = 5.84 \cdot 10^{-7}$, F-test of equality of variance), but highly similar between NR and inducer ($p = 0.56$, F-test of equality of variance); and RPE and responder ($p = 0.21$, F-test of equality of variance).

To measure branching we skeletonized the patches, and quantified the distribution of nodes per patch and condition (*Figure 3D*; *Figure 3—source data 5*). Patches in the NR and in the inducer growth mode were overwhelmingly stripe-like with no branch points (*Figure 3D*; inset I), with similar node distribution ($p = 0.64$, Wilcoxon rank sum test). In contrast, both NR and inducer differed significantly at the 0.05 level from the distribution in the RPE and responder growth mode (NR-RPE: $p = 3.93 \cdot 10^{-6}$; NR-responder: $p = 3.26 \cdot 10^{-4}$; inducer-RPE: $p = 6.24 \cdot 10^{-7}$; inducer-responder: $p = 7.00 \cdot 10^{-5}$, Wilcoxon rank sum test). Patches in the RPE and in the responder growth mode frequently bifurcated or merged, creating branching shapes with inclusions indicative of clone intermingling (*Figure 3D*; inset III). RPE and responder growth mode were highly similar in this metric ($p = 0.38$, Wilcoxon rank sum test).

Not all patches were contiguous with the embryonic retina. Such 'late arising patches' result if a cell divided intermittently with periods of dormancy, leaving clone fragments behind (*Figure 3B*, highly variable scenario). We quantified fragmentation by plotting the occurrence of late arising patches along the normalized post-embryonic retinal radius (*Figure 3E*; *Figure 3—source data 6*). In the NR late patches clustered in the central post-embryonic retina and waned thereafter. Thus clone fragments were not equally distributed, consistent with lower levels of cell division variability and a majority of continuous stripe-like clones. In contrast, the RPE displayed an even distribution indicative of frequent fragmentation throughout the life of the animal as predicted for the highly

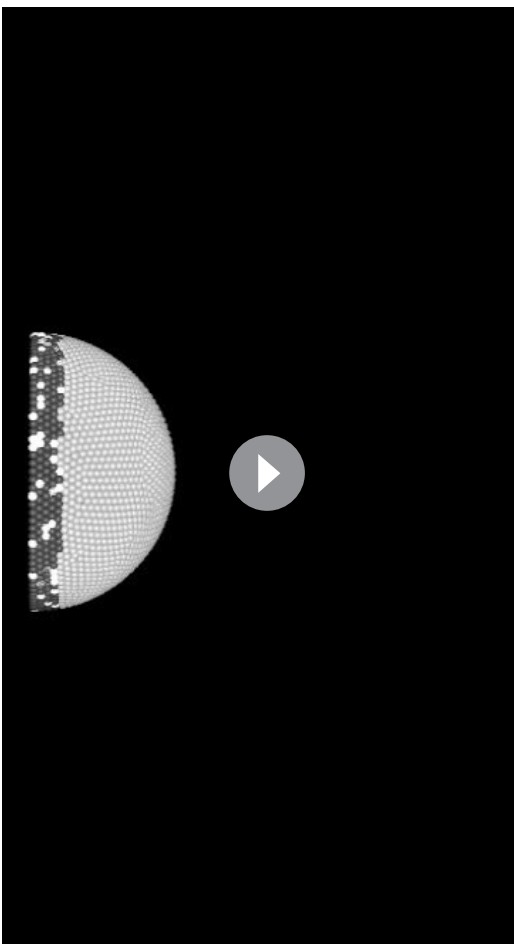

**Video 3.** Simulation where 20% of stem cells were labelled in white showing clone fusion and fragmentation. Cells that were initially differentiated are shown in light gray. When cells divide they briefly flash white.

DOI: https://doi.org/10.7554/eLife.42646.025

variable scenario (NR-RPE: p=1.74·$10^{-3}$, Wilcoxon rank sum test). The simulated data showed the same tendency, to a lesser degree, as the central peak in late patches was higher in the inducer growth mode and peripheral late patches occurred more frequently in the responder growth mode (*Figure 3E*; inducer-responder: p=0.10, Wilcoxon rank sum test). In this metric, the RPE stood out from the NR and both simulated conditions (RPE-inducer: p=6.94·$10^{-5}$; RPE-responder: p=0.04, Wilcoxon rank sum test), indicating a high degree of fragmentation and thus cell division variability.

Together, these data show that NR and RPE have different degrees of variability in cell division timing. The NR displayed lower variability consistent with the simulated inducer growth mode, while the RPE showed higher levels of variability that even exceeded what we modelled with the responder growth mode. Thus, our data support a model where NR and RPE concertedly expand relying on different growth modes, which manifest in differently shaped ArCoS.

## Stem- and progenitor cell domains are an emergent property of the system

Both the NR and simulations displayed a cluster of late patches in the central post-embryonic retina (*Figure 3E*). Additionally, when discounting late patches, the distribution of patch length showed clear bimodality (*Figure 3—figure supplement 2B*), suggesting that beyond fragmentation an additional stochastic process took place after clonal labelling. The region at the border to the embryonic retina, the 'induction ring', marks the original position of the CMZ at the timepoint of Cre-mediated recombination (*Figure 4E*). To investigate the stem cell dynamics in the induction ring we turned to the simulation.

Surprisingly, the virtual induction ring contained many few-cell clones unrelated to any ArCoS (*Figure 4A'*, encircled by pink dashed lines). In these clones, all stem cells left the niche and thus differentiated ('terminated clones'). Nested inductions showed that sister stem cells within one clone segregated into subclones (*Figure 4A'–A''*, highlighted ArCoS). However, only some of these subclones generated virtual ArCoS. Again, terminated clones clustered in the virtual induction ring (*Figure 4A''*, encircled by black dashed lines), demonstrating that the pattern repeated itself regardless of the timepoint of virtual induction. Therefore, since central positions were occupied by short terminated clones, many stripe-like patches necessarily began in more peripheral positions, explaining the peak in late arising patches.

In our model, all proliferative cells were equipotent stem cells. Nevertheless, a subset of these virtual stem cells proliferated only a few times before terminally differentiating, resulting in a bimodal distribution of patch lengths (*Figure 3—figure supplement 2B*). Notably, the overwhelming majority of virtual ArCoS emerged from the periphery of the induction ring (*Figure 4A'–A''*; *Video 4*), as confirmed by tracing back the position of the founder stem cells at simulation step 0, while centrally located cells formed exclusively terminated clones (*Figure 4B*). This behavior is highly reminiscent of retinal progenitor cells *in vivo*, which are believed to reside in the central CMZ (*Raymond et al.,*

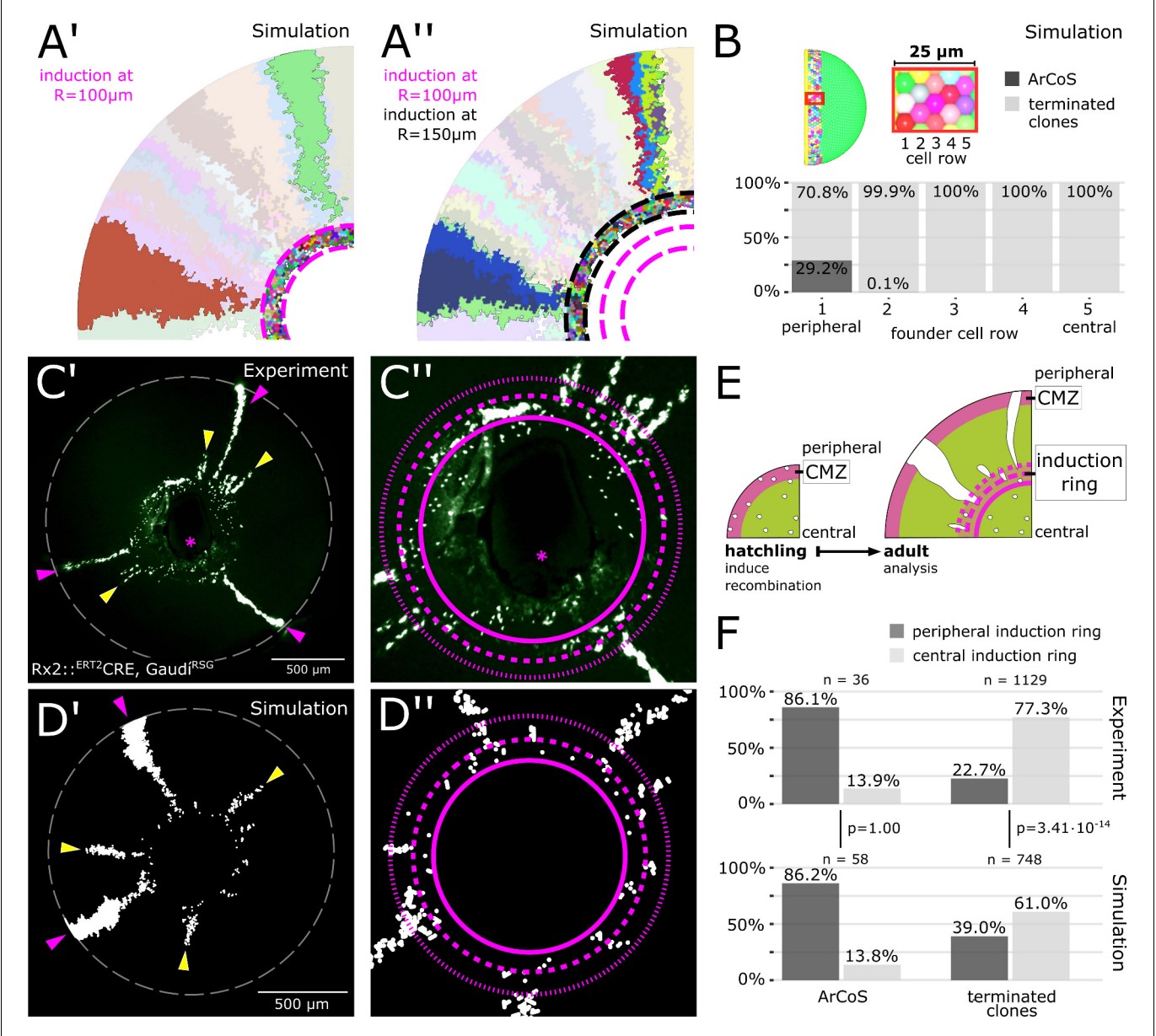

**Figure 4.** The majority of stem cells differentiates due to cell competition for niche space. (**A′**) Detail of inducer growth mode simulation where clone label was initiated at a radius of R = 100 μm. Small clusters lie centrally, while virtual ArCoS start peripherally. Two virtual ArCoS are highlighted. Pink dashed lines encircle virtual induction ring. (**A″**) Same simulation as in (**A′**), but with clone label initiated at R = 150 μm. The second wave of clonal label leads to a renewed occurrence of small clusters. Two polyclonal patches are highlighted, which correspond to subclones of the highlighted clones in (**A′**). (**B**) The majority of virtual ArCoS derives from stem cells that in simulation step 0 were located in the two most peripheral rows of the virtual CMZ. (**C′**) Proximal view of NR clones. (**C″**) Magnification of central retina from (**C′**). (**C′–C″**) Maximum projection of confocal stack of GFP signal in false colors; rotated to place optic nerve exit (pink asterisk) ventrally. (**D′**) Proximal view of simulated clones. (**D″**) Magnification of central retina from (**D′**). (**C′–D′**) Retinal edge marked by white dashed circle; dashed pink lines encircle and subdivide induction ring into central and peripheral parts; pink arrowheads mark ArCoS, yellow arrowheads mark terminated clones. (**E**) Scheme of the experiment. (**F**) Proportions of ArCoS and terminated clones arising from central and peripheral induction ring in experiment (n = 20 retinae) and simulation (n = 5 simulations, sampled six times each). p-values calculated with a 2-sample test for equality of proportions.

DOI: https://doi.org/10.7554/eLife.42646.026

The following source data and figure supplement are available for figure 4:

**Source data 1.** Origin of ArCoS and terminated clones in the simulation.
DOI: https://doi.org/10.7554/eLife.42646.028
**Source data 2.** Proportion of ArCoS and terminated clones in induction ring zones.
*Figure 4 continued on next page*

*Figure 4 continued*

DOI: https://doi.org/10.7554/eLife.42646.029

**Figure supplement 1.** Induction ring in very sparsely labelled samples.

DOI: https://doi.org/10.7554/eLife.42646.027

*2006*; *Shi et al., 2017*). Strikingly, only a minority of virtual stem cells formed ArCoS, while the vast majority formed terminated clones (*Figure 4B*).

Together, these data show that the virtual stem cell population subdivided into two functional domains that mirror the current model of the retinal niche with a peripheral stem- and a central progenitor domain (*Raymond et al., 2006*; *Shi et al., 2017*). Importantly, this subdivision was not imposed onto the simulation, but emerged dynamically. The central-most cells were poised to differentiate by being pushed out of the niche by divisions of their more peripheral neighbors. This neutral competition occurred continuously, as demonstrated by nested virtual inductions (*Figure 4A'–A''*). Thus, the spatial segregation of stem- and progenitor domains is an emergent property of the system.

## Experimental clones follow a spatially biased stochastic drift

Our simulations uncovered a role of stochastic drift in the niche, and lead us to the following two predictions: First, a large proportion of stem cells is lost by neutral competition and forms terminated clones. Thus, ArCoS should be a minority among labelled clones. Second, there is a spatial bias in this drift: The majority of ArCoS will derive from peripheral cells but some will derive from more central positions. Similarly, the majority of terminated clones will derive from central positions, but some will derive from peripheral positions.

To address these predictions experimentally, we again labelled NR stem cells in hatchlings using the Rx2::ERT2Cre, GaudiRSG line (*Centanin et al., 2014*; *Reinhardt et al., 2015*), which when recombined results in a nuclear GFP signal, and analysed the eyes at adult stage. Few-cell clusters in the induction ring vastly outnumbered ArCoS, showing that terminated clones were the most common type of clone (n = 1129 terminated clones in 20 retinae; *Figure 4C'–C''*, *Figure 4—figure supplement 1A–B*). A small fraction of terminated clones extended into the post-embryonic retina (*Figure 4C'–C''*, yellow arrowheads). ArCoS, which by definition always reach the retinal margin, were less frequent (*Figure 4C'–C''*, pink arrowheads; n = 36 ArCoS in 20 retinae). Thus, Rx2-expressing cells in the CMZ included cells that proliferated indefinitely as well as cells that proliferated only a few times before differentiating. The preponderance of terminated clones shows that ArCoS-forming cells are a minority, in line with our first prediction.

To address the spatially biased stochastic drift, we examined at which position in the induction ring clones contained their central-most pixels in experiment and simulation (*Figure 4C'–C''*, *D'–D''*, *F*). Among terminated clones, the majority started in central positions (experiment: 77.3%; simulation: 61.0%), while a minority were exclusively located in the peripheral induction ring or in the post-embryonic retina (experiment: 22.7%; simulation: 39.0%). The difference in proportions between experiment and simulation may indicate that the simulation underestimates the number of terminated

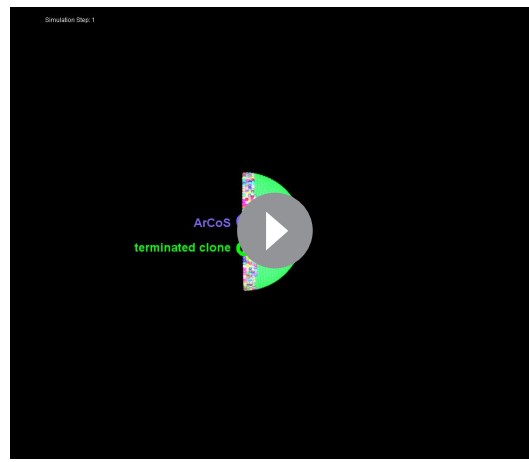

**Video 4.** A terminated clone and an ArCoS originating from the peripheral-most stem cell row. Simulation of the inducer growth mode. Two cells are highlighted in the first simulation step: A purple cell that will give rise to an ArCoS (purple circle), and a green cell that will divide only a few times before its lineage completely exits the niche, forming a terminated clone (green circle). Note how almost all proliferative cells not at the very edge of the hemisphere are pushed out of the proliferative domain and form terminated clones.

DOI: https://doi.org/10.7554/eLife.42646.030

clones. Nevertheless, a sizeable subset of experimental terminated clones derived from the periphery of the stem cell domain of the CMZ, indicating that some stem cells drifted into a progenitor-like state.

Among experimental ArCoS, the vast majority (86.1%) started in the periphery, but 13.9% derived from a central position, showing that some cells located in the central progenitor domain of the CMZ drifted into a lifelong stem cell fate. Strikingly, the ratios for peripheral and central ArCoS in the simulation are nearly identical (p=1.00, 2-sample test for equality of proportions), showing that the simulation captures ArCoS dynamics extremely well. Together, these data support a model of stochastic drift with a peripheral-stem and central-progenitor bias that is conditioned by the physical topology of the niche.

## NR stem cells undergo radial divisions at the rate predicted by shape regulation

NR ArCoS formed stripes that appeared slightly narrower than in the simulation (*Figure 3A'–A'', C'–C''*). In simulations, the division axis was not oriented ('random division axis'). The thin clonal stripes suggested that NR stem cells had a preferential axis of division along the radial (central-peripheral) coordinate, while circumferential divisions occurred with lower frequency than expected for a random division axis orientation.

We wondered whether NR stem cell division orientation could relate to shaping the organ. An inducer growth mode does not necessarily imply regulation of organ shape. To use an analogy, a mass of dough grows from within (similar to the inducer growth mode), but its shape can be imposed externally by a mold (*i.e.* the dough does not affect shape regulation). In the NR, the shape could plausibly be imposed externally by any of the surrounding tissues, and in this case, it would have no role in organ shape regulation (*Figure 5A*). As the space available for cells is imposed externally, any orientation of division axes is theoretically possible; after division cells will locally shift to optimally fill space. In an alternative scenario, organ shape could be regulated by oriented cell divisions of CMZ stem cells (*Figure 5B'*). In this scenario, a precise orientation of division axes is necessary.

We calculated the ideal proportion of circumferential and radial divisions required to maintain hemispherical geometry. We assumed two principal axes of division, and that each new cell contributed either to the area of the CMZ or to the rest of the eye (*Figure 5B''*). Circumferential divisions (two daughter cells stay in the CMZ) must be balanced by radial divisions (one daughter cell is poised to leave the niche and differentiate). A hemispherical eye of radius $R$ has the area

$$A_{\text{eye}} = 2\pi R^2, \tag{1}$$

while the CMZ forms a band of width $w$ at the base of the eye with area

$$A_{\text{CMZ}} = 2\pi R w. \tag{2}$$

Thus, we obtain an ideal ratio of circumferential to radial divisions of

$$1 : \frac{A_{\text{eye}} - A_{\text{CMZ}}}{A_{\text{CMZ}}},$$

$$1 : \frac{R - w}{w}, \tag{3}$$

that is for every one circumferential division, there must be $\frac{R-w}{w}$ radial divisions. Since $R \gg w$, radial divisions must be more frequent than circumferential divisions, and the frequency of radial divisions increases as the retinal radius grows.

To quantify circumferential stem cell divisions in experimental and simulated data, we took advantage of the exquisite temporal order of NR growth to measure ArCoS width – a proxy for circumferential stem cell divisions. To this end, we developed a pipeline that unrolled the retina as described before, and measured the number of pixels along each radial position normalized by the total circumference – effectively the angle enclosed by two rays traversing the center of the embryonic retina and the clone boundaries at every radial position (*Figure 5D''*). To only include lifelong stem

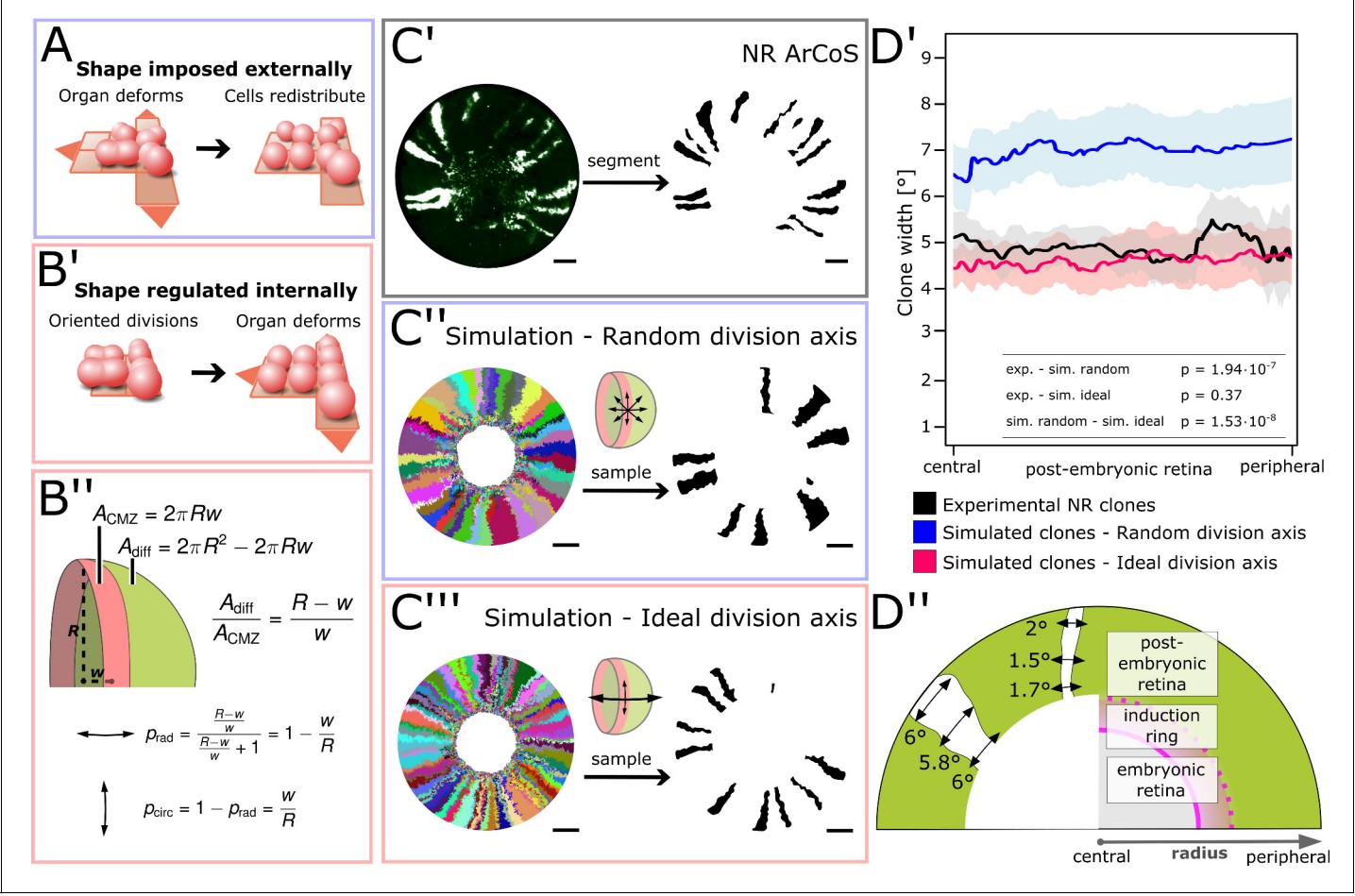

**Figure 5.** NR stem cells undergo predominant radial divisions as predicted for a shape-giving function. (**A**) If organ shape is imposed externally, then cells in the tissue will distribute to fill the available space. Regardless of cell division axes, organ geometry will lead to a directional growth in stripes. (**B'**) If organ shape is regulated by cell division axes, then oriented divisions are required. (**B''**) If the NR regulates shape through cell divisions, then more divisions along the radial axis are needed to maintain hemispherical geometry. (**C'–C''**) Examples of experimental and simulated data. For simulations, the full clone population and a random sample are shown. The initial model label was induced at R = 150 µm to match the experimental induction radius. Scale bars: 200 µm. (**D'**) Mean clone width (solid lines) and 95% confidence intervals (shaded) plotted along the post-embryonic retinal radius. Experimental data: n = 99 ArCoS across seven retinae. Simulation, random division axis: n = 102 ArCoS from five simulations; ideal division axis: n = 133 ArCoS from five simulations. p values were calculated with Welch two sample t-test. (**D''**) Schematic of radial compartments of the NR and measurements of clone width in proximal view. The clone width plotted in D' corresponds to the angle enclosed by the clone borders at every radial position.

DOI: https://doi.org/10.7554/eLife.42646.031

The following source data and figure supplements are available for figure 5:

**Source data 1.** Mean and 95% confidence interval of clone width.
DOI: https://doi.org/10.7554/eLife.42646.033

**Figure supplement 1.** Average clone width increases with increasing circumferential divisions.
DOI: https://doi.org/10.7554/eLife.42646.032

**Figure supplement 1—source data 1.** Clone width in simulations with varying circumferential bias.
DOI: https://doi.org/10.7554/eLife.42646.034

cells, we focused our analysis on the post-embryonic retina and excluded the central portion including the induction ring.

As expected, with increasing probability to divide along the circumferential axis, average clone width increases in the simulation (*Figure 5—figure supplement 1A'–B*). When division axes perfectly match the ratio in *Equation 3*, the simulation becomes the limiting case of shape regulation where the hemispherical shape is always maintained. Thus, we modelled how the 'ideal division axis' ratio

given by *Equation 3* affected simulated ArCoS in the inducer growth mode and compared this to experimental data as well as simulations with random division axis (*Figure 5C'–C'''*).

Experimental ArCoS width averaged to 4.87° (*Figure 5D'* black graph; n = 99 ArCoS across seven retinae). In contrast to experimental data, ArCoS width in simulations with random division axis averaged to 7.28° (*Figure 5D'* blue graph; n = 102 clones from five simulation runs; compared to experimental data: p=1.94$\cdot$10$^{-7}$, Welch two-sample t-test). In simulations with ideal division axis, ArCoS width closely matched experimental data, averaging at 4.54° (*Figure 5D'*, red graph; n = 133 clones from five simulation runs; compared to experimental data: p=0.37, Welch two-sample t-test).

These data show that NR stem cell divisions were not randomly oriented, but instead were preferentially oriented along the central-peripheral axis. Moreover NR stem cells underwent radial and circumferential divisions at a rate consistent with a role in organ shape regulation.

## Local biases in ventral NR stem cell divisions influence retinal topology

We observed that in the retina of the surface-dwelling medaka, the position of the embryonic retina was not centered, but instead was shifted ventrally (*Figure 6A'*). As a result, the post-embryonic retina was longer dorsally than ventrally (ratio dorsal to ventral length: mean = 1.42; standard deviation = 0.29; n = 10 retinae). The embryonic retina covered the entire retinal surface at induction (*Figure 6A''*). Equal growth around the circumference should maintain the embryonic retina in the center. The ventral-ward shift indicated that along the CMZ circumference, ventral stem cells had different division parameters.

We probed the feasibility of different scenarios in generating a ventral shift in an *in silico* screen. First, we discerned two ways for stem cells in the ventral domain (defined as a 90° sector; *Figure 6— figure supplement 2*) to select a different division behavior: Either a lineage-bound intrinsic signal (*e.g.* epigenetic imprinting), or a lineage-independent extrinsic signal (*e.g.* a local diffusible molecule). Second, we altered two cell division parameters: The probability of division, which we varied between half ($p_{\text{div\_ventral}} = 0.5 \cdot p_{\text{div\_non-ventral}}$) or equal to the value in the non-ventral sector ($p_{\text{div\_ventral}} = p_{\text{div\_non-ventral}}$), and the preferential axis of cell division, which we varied between circumferentially-biased ($p_{\text{circ\_ventral}} = 1$) and radially-biased ($p_{\text{circ\_ventral}} = 0$).

In control simulations where all cells behaved equally, the embryonic retina stayed centered (*Figure 6B', C'*). For a lineage-bound intrinsic signal, a circumferential bias lead to massive enlargement of ventral lineages at the expense of adjacent clones without affecting the embryonic retina (*Figure 6B''*). Reducing proliferation probability resulted in termination of ventral lineages, as adjacent clones displaced them from the virtual niche (*Figure 6B'''*). An intrinsic signal resulted in a ventral shift only if circumferential bias was combined with lower proliferation probability (*Figure 6B''''* – condition I). In these simulations, circumferential divisions allowed ventral lineages to physically occupy niche positions (preventing their displacement) while lower proliferation reduced pressure on cells of the embryonic retina, allowing a ventral shift. In the scenario of a lineage-independent extrinsic signal, two conditions resulted in a ventral shift of the embryonic retina: Both lower division probability (*Figure 6C'''* – condition II) and the combination of lower division probability with circumferential division axis bias (*Figure 6C''''* – condition III).

To identify which scenario was most plausible, we analysed patches in the ventral and non-ventral sectors. Both in experiments and all three simulated conditions, patch shape in the non-ventral sector was similar (*Figure 6D'–D''''*). Although there was a tendency for ventral clones to terminate more often, the width distribution of experimental NR patches did not differ substantially between non-ventral and ventral sectors (*Figure 6D', E'*, *Figure 6—figure supplement 1D'*; p=0.84, Wilcoxon rank sum test). In contrast, this latter criterion was violated by two of the three simulated scenarios (*Figure 6D''–D'''' and E''–E''''*, *Figure 6—figure supplement 1D''-D''''*).

In condition I, ventral ArCoS started narrow but then broadened (*Figure 6E''*) and interdigitated circumferentially (*Figure 6—figure supplement 1A*, black arrowheads), unlike the very uniform stripes in the experimental data. The broader ventral ArCoS lead to a more dispersed distribution compared to the non-ventral sector (*Figure 6—figure supplement 1D''*; p=4.31$\cdot$10$^{-14}$, Wilcoxon rank sum test). In condition II, the majority of ventral ArCoS formed very narrow stripes, but at the border to the non-ventral sector ArCoS were broad and curved (*Figure 6—figure supplement 1B*, black arrowheads). Again, this resulted in more shape variation (*Figure 6E'''*). Nevertheless, these outliers were outweighed by a high density of narrow clones, such that the overall distribution was similar between ventral and non-ventral sectors (*Figure 6—figure supplement 1D'''*; p=0.12,

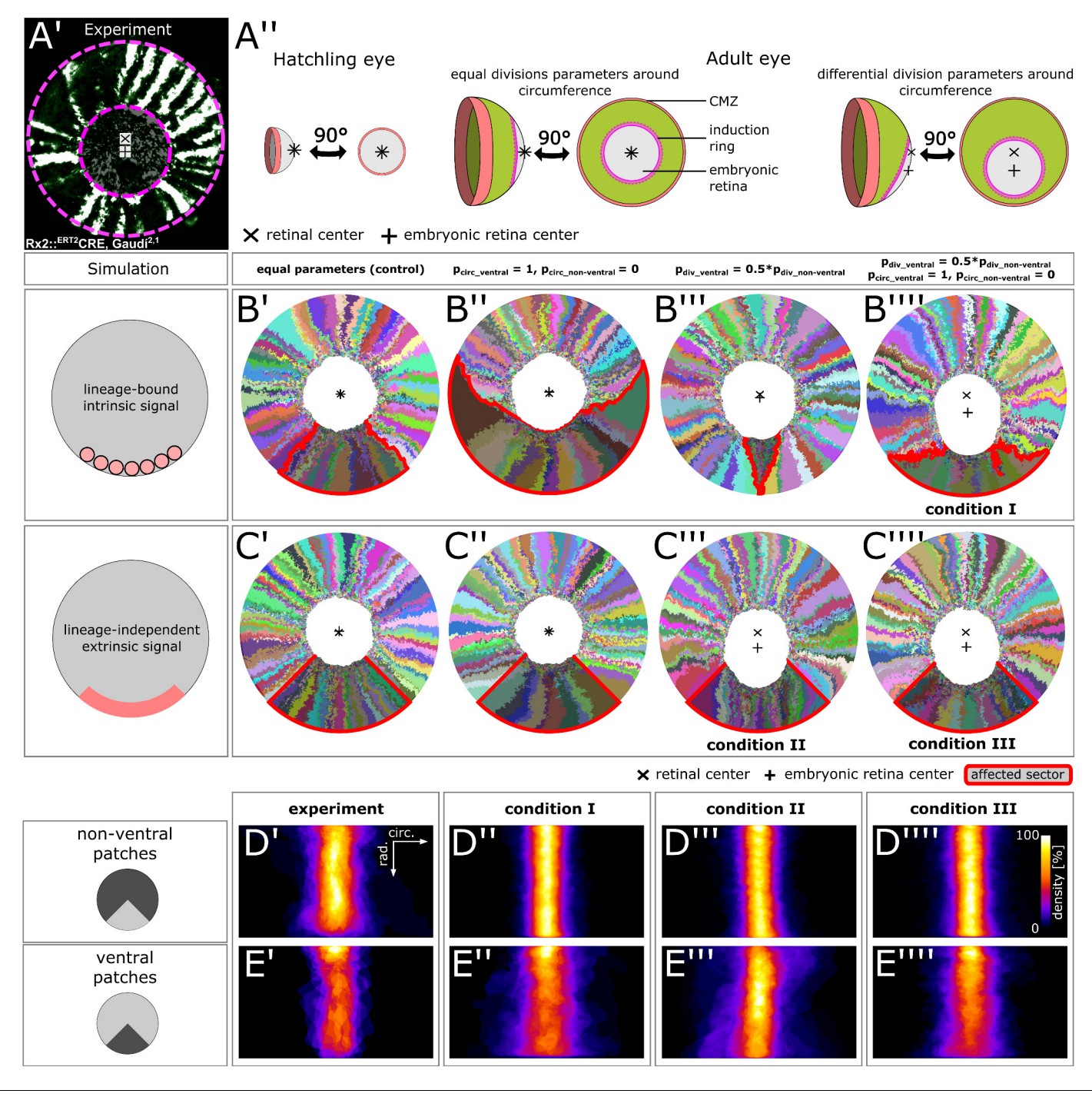

**Figure 6.** Stem cells in the ventral CMZ have different proliferation parameters. (**A'**) Proximal view of NR clones highlighting the discrepancy between retinal center and embryonic retinal center. Depicted sample is the same as in *Figure 1D*. (**A''**) A differential proliferative behavior along the CMZ circumference can explain the shift in position of the embryonic retina. (**B'–B''''**) Simulations where lineages whose embryonic origin is in the ventral sector inherit a signal that leads to different proliferation parameters. A shift occurred when ventral lineages had both lower division probability and higher circumferential divisions. Clones originating in ventral embryonic CMZ are outlined in red. (**C'–C''''**) Simulations where all cells in a ventral 90° sector exhibit different proliferation parameters regardless of lineage relationships. A shift occurred in conditions with slower proliferation as well as slower proliferation combined with circumferential division axis bias. (**D'–E''''**) Patch superposition for experimental data as well as the three simulated conditions that display a ventral shift of the embryonic retina. (**D'–D''''**) Non-ventral patches. (**E'–E''''**) Ventral patches.

DOI: https://doi.org/10.7554/eLife.42646.035

The following source data and figure supplements are available for figure 6:

*Figure 6 continued on next page*

*Figure 6 continued*

**Source data 1.** Patch outlines.
DOI: https://doi.org/10.7554/eLife.42646.038
**Figure supplement 1.** Magnification of simulations displaying a ventral shift.
DOI: https://doi.org/10.7554/eLife.42646.036
**Figure supplement 1—source data 1.** Patch width distribution.
DOI: https://doi.org/10.7554/eLife.42646.039
**Figure supplement 2.** Definition of the ventral sector in the simulation.
DOI: https://doi.org/10.7554/eLife.42646.037

Wilcoxon rank sum test). Clones in the ventral and non-ventral sectors were qualitatively similar in condition III (*Figure 6E''''*, *Figure 6—figure supplement 1C*). Ventral clones however tended to be broader, resulting in a more dispersed distribution compared to the non-ventral sector (*Figure 6—figure supplement 1D''''*; p=$7.29 \cdot 10^{-7}$, Wilcoxon rank sum test).

In conclusion, ventral NR stem cells have a different behavior than elsewhere along the circumference, leading to a ventral-ward shift of the embryonic retina. The simulations suggest that this different behavior consists of modulation of proliferation parameters by an extrinsic signal in the ventral CMZ.

## Discussion

### The NR drives growth upstream of the RPE

The coordinated growth of multiple independent tissues is a ubiquitous process in biology. In this work, we used the post-embryonic growth of NR and RPE in the eye of medaka as a model system of coordination in an organ where both growth and shape must be precisely regulated.

Eye size in fish scales to the body size (*Lyall, 1957*; *Johns and Easter, 1977*). Body size, and thus eye growth rates greatly vary among individuals and depend on environmental factors (*Johns, 1981*). This natural malleability implies that feedback coupling plays a dominant role rather than the precise parametrization of each tissue growth and cell proliferation rate. Our simulations showed that inducer and responder growth modes impacted on variability in cell division timing, ultimately resulting in distinct clonal patterns that reproduced the experimentally observed differences between NR and RPE.

RPE cells divided with high variability, indicative of periods of long quiescence where they waited for proliferative cues. NR cells displayed lower variability, supporting an upstream role in regulating growth (*Figure 7A*). Although our implementation of the responder growth mode used a mechanical stimulus (local cell density), a biochemical stimulus could equally well represent the system.

Our model highlights an underappreciated mechanism whereby tissues coordinate by inducer and responder roles. Such division of labor among tissues might apply more generally to multiple organ systems, for example hair follicle cells in mouse induce the growth of underlying adipose tissue through hedgehog signalling (*Zhang et al., 2016*). Intriguingly, hedgehog signalling also regulates the NR/RPE boundary in the CMZ of medaka (*Reinhardt et al., 2015*), suggesting that signals mediating coordination of proliferative cell populations might be conserved.

### Multipotent progenitor cells are stem cells that were outcompeted

The topology of the retinal niche lead to a spatially biased neutral drift where stem and progenitor compartments spontaneously emerged. All virtual cells had equal potency, yet only a fraction realized their full stem cell potential. Peripheral cells had a high chance to become canalized in a stem cell fate, while central cells were more likely to act as progenitor cells with limited proliferation potential (*Figure 7B*).

Our experimental data support a spatially biased neutral drift. Fusion of clones may have lead us to overestimate ArCoS deriving from the central domain, which represent progenitors reverting to a stem cell fate. Nevertheless, terminated clones arising from the very periphery of the niche unambiguously demonstrate that some stem cells failed to self-renew throughout the life of the animal. Moreover, our finding that only cells in the first two rows of the CMZ have stem cell potential is consistent

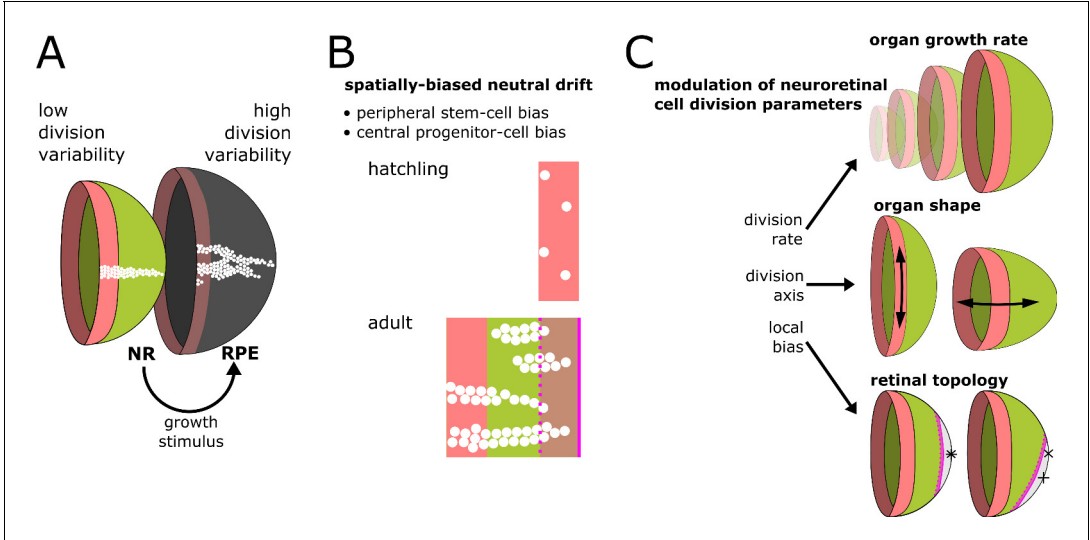

**Figure 7.** Summary of results and proposed model of CMZ dynamics. (**A**) Growth coordination of NR and RPE is achieved by the NR providing instructive stimuli that modulate proliferation of RPE stem cells. As a result of the different growth strategies, variability in cell division timing is elevated in the RPE and lowered in the NR. (**B**) A base level of variability persists in the NR, such that individual stem cells may differentiate and some multipotent progenitor cells drift to a stem cell fate according to a spatially biased neutral drift model. Thus, stem cells and multipotent progenitor cells have identical proliferative potency. (**C**) Schematic summary of findings and proposed model, where different NR cell proliferation parameters affect both global and local retinal properties.

DOI: https://doi.org/10.7554/eLife.42646.040

with *in vivo* time-lapse data (*Wan et al., 2016*; *Tang et al., 2017*). Interestingly, retrograde movement of row 2 cells into row 1 of the CMZ occurs *in vivo* (*Wan et al., 2016*), which we also observed in our simulations.

CMZ progenitor cells can be subdivided into two populations (*Harris and Perron, 1998*; *Raymond et al., 2006*): First, peripheral multipotent progenitors (*i.e.* able to generate all retinal neurons and glia) which differ from stem cells only in their proliferative potential. Second, central progenitors that are restricted both in proliferative and differentiation potential, which likely act as a transit-amplifying zone, both increasing the proliferative output and cross-regulating to produce a full neuronal complement with the correct proportions of cell types (*Pérez Saturnino et al., 2018*).

Our data support an alternative model that identifies peripheral multipotent progenitors as stem cells that have been outcompeted. All terminated clones we examined were multipotent and spanned all retinal layers (*Figure 1—figure supplement 2*). Thus, as in many other systems (*Clevers and Watt, 2018*), our work highlights the limitation of strictly defining stem cells as infinitely self-renewing, or *a posteriori* based on their ArCoS-forming capacity.

Importantly, although stochastic competition is most apparent in the early phase after clonal induction, it occurs continuously as demonstrated by late arising patches (*Figure 3E*) and nested inductions (*Figure 4A'–A''*). The shift from an 'early stochastic' to 'late polyclonal' growth observed in other systems (*Nguyen et al., 2017*) may simply result from clonal growth masking the underlying stochasticity. Due to this stochasticity, it is impossible to tell at any moment with absolute certainty if a given cell will perpetually function as a stem cell.

## Why does the CMZ niche of the retina not drift to monoclonality?

Neutral drift in a finite-sized environment such as adult mammalian tissues must ultimately result in a monoclonal niche (*Snippert et al., 2010*; *Colom and Jones, 2016*; *Clevers and Watt, 2018*). In fish, homeostatic growth expands niches, and thus the number of stem cells increases (*Centanin et al., 2011*). In principle, niche expansion reduces the impact of competition on clonal loss, but does not completely abolish it.

Indeed, neutral drift leads to gradual loss of polyclonality in the intestine and muscle of fish (*Aghaallaei et al., 2016*; *Nguyen et al., 2017*). Organs may limit monoclonal drift by physically

isolating niches (*Aghaallaei et al., 2016*). In the intestine of both mammals and fish, physical isolation of multiple niches results in a polyclonal organ built up of monoclonal units (*Snippert et al., 2010*; *Aghaallaei et al., 2016*). In contrast, the CMZ is a physically contiguous niche that nevertheless maintains polyclonality lifelong both in the NR and the RPE (*Centanin et al., 2011*; *Centanin et al., 2014*). As shown in this work, the retina is not devoid of stochastic competition. Then how does it conserve its polyclonality?

Conceptually, the clonal growth of the retina resembles a population expanding into a new habitat, as studied in the context of evolutionary theory (*Hallatschek and Nelson, 2010*). Specifically for a radially expanding population, it has been mathematically proven that (assuming pure neutral genetic drift) no single clone will ever take over and clonal sectors perpetually coexist (*Hallatschek and Nelson, 2010*; *Korolev et al., 2012*). Growth of the perimeter is faster than circumferential expansion of clones, thus preserving population diversity (*Hallatschek and Nelson, 2010*). Interestingly, in the NR, the biased division axis further reduces competition (*Figure 5*), thus increasing niche polyclonality. In summary, the geometry of the CMZ niche prohibits the total loss of polyclonality.

## The NR senses the retinal radius and directs cell divisions to adapt organ shape

Our analysis of NR cell divisions implies that cells sense the radius of the eye to regulate organ shape. Across vertebrates, the retina integrates visual input to adapt organ shape to optimize optics, a process called 'emmetropization' (*Wallman and Winawer, 2004*). In chicken, emmetropization is regulated by specialized neurons distributed across the retina that send their axons to the CMZ, implicating the CMZ in regulation of eye shape (*Fischer et al., 2008*). Visual cues also guide emmetropization in fish (*Kröger and Wagner, 1996*; *Shen et al., 2005*; *Shen and Sivak, 2007*). Eye growth in young fish predominantly occurs by cell addition, while in older fish CMZ proliferation decreases (*Johns, 1981*) coincident with a decrease in emmetropization plasticity (*Shen and Sivak, 2007*). Thus, in fish, emmetropization correlates with CMZ proliferation.

Experiments in chicken and zebrafish support the existence of two principal axes of stem cell division, that is circumferential and central-peripheral (*Fischer et al., 2008*; *Ritchey et al., 2012*; *Wan et al., 2016*). Notably, the predominance of central-peripheral divisions and decreasing frequency over time of circumferential divisions in CMZ stem cells that is predicted by *Equation 3* is supported by *in vivo* imaging data (*Wan et al., 2016*) and previous long-term clonal analyses (*Centanin et al., 2014*). Altogether, the data support a model where the NR perceives the retinal radius through visual cues, and that cell divisions in the NR contribute to shaping the eye.

## An eye-internal signal directs local proliferation parameters in the CMZ

The retinae of many fishes grow asymmetrically, perhaps to maintain the relative positions of receptive fields of neurons (*Johns, 1977*; *Johns, 1981*; *Easter, 1992*). Ecology dictates a distribution of subdomains enriched in specialized neuronal circuits and retinal cell subtypes (*Zimmermann et al., 2018*). Interestingly, in green sunfish, the area that grows slowest displays highest visual acuity (*Cameron, 1995*). Medaka predominantly gaze upwards in their native shallow rice paddies, and a higher ventral acuity has been presumed based on photoreceptor densities (*Nishiwaki et al., 1997*). Thus, slower ventral growth may have evolved to match ecological requirements for medaka vision.

Our *in silico* screen identified three scenarios consistent with asymmetric ventral growth. Based on clonal patterns, an extrinsic signal driving lower proliferation (and potentially also circumferential divisions) appears most plausible. Experimental eye re-orientation *in vivo* implied an eye-internal mechanism independent on body axes or visual cues in regulating asymmetric retinal growth (*Cameron, 1996*). The origin of this signal and how it scales with the growing eye to always affect a similarly-sized retinal sector remains to be elucidated.

## The CMZ integrates cues to direct eye growth and shape

The retina integrates global systemic cues such as nutrition to scale with body size (*Johns and Easter, 1977*), local eye-internal cues to generate an asymmetric retinal topology (*Cameron, 1996*), and external visual cues to adapt the shape of the organ (*Kröger and Wagner, 1996*; *Shen and Sivak, 2007*). In chicken and goldfish, visual cues and nutrients feed into the CMZ through growth factor

signalling (*Boucher and Hitchcock, 1998*; *Fischer et al., 2008*; *Ritchey et al., 2012*). We propose that NR cells in the CMZ act as a hub to coordinate organ growth; in the eye of fish, this happens at the level of cell proliferation parameters, which affect eye growth, eye shape, and retinal topology (*Figure 7C*).

Indeterminate, lifelong growth is a widespread evolutionary strategy (*Karkach, 2006*). Given the geometrical constraints of the eye with respect to optics, a peripheral proliferative domain is the most parsimonious architecture to ensure that the differentiated neuronal cell mosaic is not disturbed by constant proliferation. Fishes are the largest vertebrate clade, with a huge diversity of eye shapes, such as cylindrical eyes in deep-sea fish (*Fernald, 1990*). By modulating CMZ proliferation parameters, evolution can adapt whole-organ morphogenesis to perfectly fit to the species' ecological niche.

# Materials and methods

## Key resources table

| Reagent type (species) or resource | Designation | Source or reference | Identifiers | Additional information |
|---|---|---|---|---|
| Strain (*Oryzias latipes*) | Cab | *Loosli et al., 2000* | | wildtype inbred strain derived fom wild medaka Southern population |
| Strain (*O. latipes*) | Rx2::$^{ERT2}$Cre, Gaudi$^{RSG}$ | *Centanin et al., 2014*; *Reinhardt et al., 2015* | | |
| Strain (*O. latipes*) | Rx2::$^{ERT2}$Cre, Gaudi$^{2.1}$ | *Centanin et al., 2014*; *Reinhardt et al., 2015* | | |
| Strain (*O. latipes*) | Gaudi$^{LoxP-OUT}$ | *Centanin et al., 2014* | | Derived from recombined gametes of Ubi::Gaudi$^{RSG}$. |
| Sequence-based reagent | short guide RNAs against Oca2 | this paper and *Lischik et al., 2019* | | target sites: GAAACCCAGGTGGCCATTGC[AGG] and TTGCAGGAATCATTCTGTGT[GGG] |
| Chemical compound, drug | Tamoxifen | Sigma Aldrich | T5648 | |
| Chemical compound, drug | 5-Iodo-2'-deoxyuridine (IdU) | Sigma Aldrich | I7756 | |
| Chemical compound, drug | Ethyl 3-aminobenzoate methanesulfonate salt (Tricaine) | Sigma Aldrich | A5040 | |
| Antibody | anti-GFP (chicken, polyclonal) | Life Technologies | A10262 | 1:200 |
| Antibody | anti-IdU (mouse, monoclonal) | Becton Dickinson | 347580 | 1:25 |
| Antibody | anti-chicken Alexa Fluor 488 (donkey, polyclonal) | Jackson/Dianova | 703-545-155 | 1:200 |
| Antibody | anti-mouse Alexa 546 (goat, polyclonal) | Invitrogen | A-11030 | 1:400 |

## Experimental methods

### Animal welfare statement

Medaka (*Oryzias latipes*) fish were bred and maintained as previously established (*Loosli et al., 2000*). All experimental procedures were performed according to the guidelines of the German animal welfare law and approved by the local government (Tierschutzgesetz §11, Abs. 1, Nr. 1, husbandry permit number AZ 35–9185.64/BH; line generation permit number AZ 35–9185.81/G-145–15).

## Clonal lineage labelling

ArCoS in the NR were generated as described previously (*Centanin et al., 2011*; *Centanin et al., 2014*; *Reinhardt et al., 2015*). Transplantations were from labelled donor cells of the LoxP$^{OUT}$ line to unlabelled wildtype Cab host blastulae. Cre-mediated recombination was performed in hatchlings by induction of the Rx2::$^{ERT2}$Cre, Gaudí lines with 5 µM tamoxifen diluted in fish water for at least 3 hr.

For ArCoS in the RPE, mosaic unpigmented clones were generated using CRISPR/Cas9 by injecting 150 ng/µl Cas9 mRNA and 30 ng/µl each of two short guide RNAs directed against the gene Oca2 in one-cell stage Cab medaka embryos. Oca2 is required to produce melanin pigment (*Fukamachi et al., 2004*; *Lischik et al., 2019*). The sgRNA was designed using CCTop (*Stemmer et al., 2015*).

## Treatment with IdU

Fish were bathed in fish water containing concentrations of 2.5 mM IdU as previously described (*Centanin et al., 2011*).

## Sample preparation and imaging

Fish were allowed to grow and sacrificed as young adults with an overdose of Tricaine. Whole fish were fixed in 4% formaldehyde in phosphate buffered saline and 0.1% Tween (PTW) at least once overnight at 4°C while gently shaking. Eyes were dissected, if necessary immunostained, and imaged at a Nikon AZ100 upright stereomicroscope using a 5x dry objective.

## Immunostaining

To remove melanin pigment, fixed samples were bleached with 0.3% $H_2O_2$ and 0.5% KOH dissolved in PTW. Samples were permeabilized in acetone for 10 min at −20°C. Blocking was performed for at least one hour in a solution of 4% sheep serum, 1% bovine serum albumin (BSA), and 0.1% DMSO, diluted in PTW. Samples were incubated with primary antibodies diluted in 4% sheep serum and 1% BSA in PTW at least once overnight at 4°C with gentle mixing. Secondary antibodies were diluted in 4% sheep serum and 1% BSA in PTW; samples were incubated in secondary antibody solution at least once overnight at 4°C with gentle mixing.

An antigen retrieval step was performed prior to IdU staining. This step consisted of post-fixation in 4% formaldehyde for 1 hr, DNA denaturation with 2M HCl and 0.5% Triton for 45 min, and pH recovery for 10 min in a 40% borax solution in PTW.

## Data analysis

### Experimental clone segmentation

All image processing and analysis was performed using the Fiji distribution of ImageJ (*Schindelin et al., 2012*). Experimental retinae were selected such that only sparsely labelled eyes of comparable size were used for analysis. For NR samples, a maximum intensity projection of confocal stacks was used for segmentation. For RPE samples, a custom script was written to create a focused reconstruction from multiple focal planes based on the hemispherical shape of the whole-mount retina. Briefly, the regions in focus in a stack through a hemispherical object are rings of increasing radii (and a circle in the first plane). The size of these rings was calculated based on the size of the sample. The focused areas were extracted and collated in one composite image.

Labelled clones were segmented by subtracting background noise with a difference of gaussians, and thresholded by the Phansalkar local threshold algorithm as it is implemented in Fiji (*Phansalkar et al., 2011*). The segmentation was manually curated to eliminate errors.

### *In silico* clonal lineage labelling

For simulating NR clones, all proliferating cells in the model received a unique identifier when the eye radius reached 150 µm. The radius was chosen based on the estimated radius of the NR when genetic recombination was induced *in vivo*. To replicate RPE clones, the virtual labelling experiment began at 100 µm, since mosaic knockout happens at an earlier timepoint in development. The

identifier was inherited to daughter cells, allowing to reconstruct a lineage at any time during the simulation.

For comparison to experimental data, between 8–13% of clones were randomly sampled from the full simulated population; the sample was chosen to produce a sparse label with a comparable number of patches per retina as in the experimental data. Each simulation was sampled twice. The sample of simulated clones was plotted as a 2D projection using a custom Python script; cellular edges were blurred by application of a median filter and shape smoothing plugin in ImageJ.

### Clone shape complexity

Shape complexity of simulated clones from simulation screenshots was quantified by thresholding individual clones by color, calculating the pixel perimeter, and dividing this value by the pixel perimeter of the smallest bounding rectangle enclosing the clone.

### Patch shape analysis

Data analysis on experimental and simulated data was performed using the same automated pipeline in ImageJ, which takes as an input segmented images where the embryonic retina and retinal margin were marked manually. The size of the embryonic retina was estimated based on the induction ring and position of the optic nerve exit, the radius of this estimate was then increased to ensure complete exclusion of all embryonic area. Different sizes of this estimate produced comparable results.

The analysis pipeline first performed a coordinate transform to unroll the retina: Proximal views of experimental and simulated retinae were centered on the embryonic retina, converted to polar coordinates, and finally projected onto a cartesian coordinate system (*Figure 3—figure supplement 2C*). After this transform, the width of the image corresponded to the circumference, while the height corresponded to the radius. Radii were normalized to extend from 0% (the border of the embryonic retina; central) to 100% (the retinal margin; peripheral). Patch outlines were automatically extracted and superimposed to generate patch density plots. The 'plot profile' function in ImageJ was used to extract average pixel intensities along a rectangle spanning the entire image. Gaussian fit was produced in R (*R Development Core Team, 2015*).

Skeletonization of patches for node counting was performed using a custom algorithm tailored to the radially oriented retinal lineages: Segmented patches were broken up into radial segments along normalized radial bins ranging from the embryonic to the retinal margin. Each segment was assigned a skeleton element, and these elements were linked in a final step prior to node counting.

For each patch, the starting position along normalized radial bins was noted. Patches that did not begin in the first radial bin were considered 'late arising patches'. Maximum patch width and maximum patch length were obtained by extracting individual patch outlines and computing the width and height of the minimum bounding rectangle, respectively. To exclude small spot-like patches, only patches spanning at least 20% of the radial coordinate were used for the maximum patch width analysis. Late arising patches were excluded from the maximum patch length analysis. These data were used to generate rug plots in R. Statistical analysis was performed in R.

### Quantification of the proportion of ArCoS and terminated clones

In simulated data, ArCoS were defined as clones that still retained cells in the virtual CMZ at the final simulation step used for analysis, that is when the virtual retina had attained a radius of R = 800 μm. All other clones counted as terminated clones. The initial position at simulation step 0 of the founder stem cells for each clone was extracted from the simulated data and assigned to a 5 μm-wide bin corresponding to each of the cell rows in the virtual CMZ.

For the comparison of experimental to simulated data, segmentation was performed as described in 'Experimental clone segmentation' and '*In silico* clonal lineage labelling'. The position of the induction ring was estimated based on the following criteria: The inner circle was placed such that it enclosed as many 1 cell clones as possible (*i.e.* labelled differentiated cells in the experimental data). The outer circle was placed such that it enclosed all few-cell clusters and crossed all ArCoS. Variation of the position of these two boundaries produced similar results. The induction circle was split in the middle and each clone was assigned to the central-most or peripheral-most ring based on the position of its central-most pixel.

## Width of clones

Both experimental and simulated data were projected onto a rectangular coordinate system as described in 'Patch shape analysis'. The width of clones was measured using a custom ImageJ plugin that measures the exact clone width in pixels at every radial coordinate, and normalizes this value to the circumference of the retina at the corresponding position. These measurements correspond to the angle enclosed by two rays traversing the center of the embryonic retina and the clone boundaries at every radial position (*Figure 5D''*). These width measurements were exported for analysis and plotting in R. To evaluate only lifelong stem cell clones, the induction ring and small clones that did not extend more than 10% of the radius past the induction ring were excluded from the analysis. Near the retinal margin, the fluorescent signal tapers off due to the retinal curvature and optical limitations of the imaging setup. Thus, the last 5% of the retinal radius were excluded from the analysis. The mean and 95% confidence interval were calculated for each radial position.

## Acknowledgments

We thank the Wittbrodt department, S Lemke, U Schwarz, I Lebovka, E Kuchen, R Hodge, and M González-Gaitán for critical reading of the manuscript. We thank T Thumberger and M Stemmer for their help in designing short guide RNAs for CRISPR/Cas9 experiments. We are grateful to A Saraceno, E Leist and M Majewski for fish husbandry.

Simulations in this work were performed on the computational resource bwUniCluster funded by the Ministry of Science, Research and Arts and the Universities of the State of Baden-Württemberg, Germany, within the framework program bwHPC.

## Additional information

### Funding

| Funder | Grant reference number | Author |
|---|---|---|
| European Union Seventh Framework Programme | ERC advanced grant GA 294354-ManISteC | Joachim Wittbrodt |
| Research Training Group Mathematical Modelling for the Quantitative Biosciences | | Niels Grabe |
| Heidelberg Biosciences International Graduate School HBIGS | MSc/PhD fellowship | Erika Tsingos |
| Joachim Herz Stiftung | Add-On Fellowship for Interdisciplinary Science | Erika Tsingos |

The funders had no role in study design, data collection and interpretation, or the decision to submit the work for publication.

### Author contributions

Erika Tsingos, Conceptualization, Data curation, Software, Formal analysis, Validation, Investigation, Visualization, Methodology, Writing—original draft, Writing—review and editing, Software for analysis and simulation, Performed and analyzed simulations; Burkhard Höckendorf, Conceptualization, Data curation, Software, Formal analysis, Investigation, Methodology, Writing—review and editing, Software for analysis; Thomas Sütterlin, Conceptualization, Software, Writing—review and editing, Software for simulation; Stephan Kirchmaier, Investigation, Writing—review and editing; Niels Grabe, Conceptualization, Supervision, Funding acquisition, Writing—review and editing; Lazaro Centanin, Conceptualization, Supervision, Investigation, Writing—review and editing; Joachim Wittbrodt, Conceptualization, Resources, Supervision, Funding acquisition, Writing—original draft, Project administration, Writing—review and editing

**Author ORCIDs**
Erika Tsingos (iD) http://orcid.org/0000-0002-7267-160X
Lazaro Centanin (iD) http://orcid.org/0000-0003-3889-4524
Joachim Wittbrodt (iD) https://orcid.org/0000-0001-8550-7377

**Ethics**
Animal experimentation: All experimental procedures were performed according to the guidelines of the German animal welfare law and approved by the local government (Tierschutzgesetz §11, Abs. 1, Nr. 1, husbandry permit number AZ 35-9185.64/BH; line generation permit number AZ 35-9185.81/G-145-15).

**Decision letter and Author response**
Decision letter https://doi.org/10.7554/eLife.42646.052
Author response https://doi.org/10.7554/eLife.42646.053

## Additional files

### Supplementary files

• Supplementary file 1. EPISIM Simulator executable model file. Compiled model file that can be opened in EPISIM Simulator to simulate the model described in this work.
DOI: https://doi.org/10.7554/eLife.42646.041

• Supplementary file 2. EPISIM Model project archive. Model project file that can be imported in EPISIM Modeller to visualize the cell behavioral logic implemented in the model described in this work.
DOI: https://doi.org/10.7554/eLife.42646.042

• Supplementary file 3. Instructions for using supplementary model files. Step by step instructions on how to open *Supplementary file 1* and *Supplementary file 2*.
DOI: https://doi.org/10.7554/eLife.42646.043

• Source code 1. EPISIM Simulator implementation of model described in Appendix 1. Excerpt of the relevant parts of EPISIM Simulator source code. The full source code is available at https://gitlab.com/EPISIM/EPISIM-Simulator (*Sütterlin, 2019*; copy archived at https://github.com/elifes-ciences-publications/EPISIM-Simulator).
DOI: https://doi.org/10.7554/eLife.42646.044

• Transparent reporting form
DOI: https://doi.org/10.7554/eLife.42646.045

### Data availability

All data generated or analysed during this study are included in the manuscript and supporting files. Source data files have been provided for Figures 3; 4; 5; 6. Model description and list of parameters are in the appendix. EPISIM Modeller project archive and EPISIM Simulator executable as well as instructions for use have been provided as supplementary files. The relevant parts of the source code containing the implementation of the model as described in the appendix have been provided as supplementary files. The full source code of EPISIM Simulator is available at: https://gitlab.com/EPISIM/EPISIM-Simulator (copy archived at https://github.com/elifesciences-publications/EPISIM-Simulator).

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

# Appendix 1

DOI: https://doi.org/10.7554/eLife.42646.046

## 1 Center-based biomechanical model

The biomechanical model governing physical interactions between cells and all associated parameter values was adapted from previous work (*Sütterlin et al., 2017*). The model is an off-lattice, center-based overlapping spheres model. Such cell center-based models allow cells to move freely in space and consider all forces as acting on a cell's center of mass. As previously described (*Sütterlin et al., 2017*), cells in a simulation equilibrate the distance to each of their adjacent neighbors by exerting pressure or adhesion forces. Essentially, cells optimize the distance to all neighbors until they reach a target distance, which is a function of the cell's radius, the neighboring cells' radius, and the optimal cell-cell overlap (chosen by parameter scan to create a densely packed cell ensemble). In the absence of proliferation, cell death, and movement, all cells will reach a stable distance equilibrium. Additionally, the availability of space for the cells to move in (*e.g.* tissue boundaries) affects the distance equilibrium.

In the model extensions developed in this work, the distance equilibrium is continuously perturbed by proliferating cells in the CMZ, and cells are allowed to move only on the hemispherical surface area of the eye globe. In the following, we explain new model elements introduced in the current work.

## 2 Implementation of the fish eye model

### 2.1 Model initialization

We implement the eye globe as a sphere with an initial radius $R_{init}$, centered at s, and constrain cells to remain on the surface of one hemisphere only. To generate the model's initial condition and achieve the initial distribution of cells on a hemisphere, we proceed in four steps.

1. Approximate the ideal number of cells $N_{init}$ that fit on the initial hemispherical area based on the overlapping spheres model:

$$N_{init} = \left\lceil \frac{R_{\mathrm{init}}^2}{(r\delta_{\mathrm{ol_{max}}})^2} \right\rceil,$$

(1)

where $\delta_{\mathrm{ol_{max}}}$ is the optimal overlap between cells (*Sütterlin et al., 2017*). *Appendix 1—equation (1)* is derived from the equation for the curved surface area of a hemisphere and the assumption that each cell occupies a circular area proportional to its radius and the optimal overlap.
2. Obtain a set of nodes by subdividing an icosahedral mesh on the sphere.
3. Place a cell $c$ on a mesh node located at $\mathbf{r}_c$ if it satisfies the condition

$$\mathbf{r}_{c_1} > \mathbf{s}_1,$$

(2)

where the subscript 1 denotes the x-component of the 3-dimensional vectors. This condition ensures that only one hemisphere is populated by cells. Step three is repeated until all $N_{\mathrm{init}}$ cells have been placed.
4. Simulate biomechanical forces using the model developed in *Sütterlin et al. (2017)* until cells reach equilibrium, which is defined by the average displacement $\Delta\mathbf{r}$ of all cells falling under a threshold $\mu$ during one step $\Delta t$ of the biomechanical model calculation

$$\frac{1}{N_{\mathrm{init}}} \sum_{i=1}^{N_{\mathrm{init}}} \Delta\mathbf{r}_i < \mu.$$

(3)

## 2.2 Constraining cells to the hemisphere's surface

Since the eye grows slowly over a period of several months, the growth process can be considered quasi-static. The tissues we model in this work consist of hemispherically arranged cell ensembles without mixing of cells along the direction normal to their hemispherical layer. In the center-based overlapping spheres model, the balance of forces can result in cell displacement in any direction. To restrict movement along the normal direction, we reposition cells at the end of each biomechanical force calculation step $\Delta t$ (**Figure 2—figure supplement 2**). The force balance is then iteratively recalculated with the new cell position, allowing the cell ensemble in the simulation to reach a distance equilibrium on the curved hemispherical surface. In total, each simulation step consists of 100 such iterations.

For a cell $c$ at $\mathbf{r}_c$, we obtain the new location $\tilde{\mathbf{r}}_c$ by rescaling the unit vector from the hemisphere's center s to $\mathbf{r}_c$ with the eye radius at a given simulation step $R(t)$

$$\tilde{\mathbf{r}}_c = \frac{\mathbf{s} - \mathbf{r}_c}{\|\mathbf{s} - \mathbf{r}_c\|} R(t). \tag{4}$$

To constrain cells to one hemisphere only, we introduce a ring of tightly packed immobile 'obstacle cells' on the sphere's equator that produce a biomechanical roadblock and do not otherwise participate in the simulation (**Figure 2—figure supplement 3**). Force balance between cells in the simulation and obstacle cells is calculated without using the adhesive term (**Sütterlin et al., 2017**).

## 2.3 Growth of the eye globe

Growth of the eye globe is achieved by increasing its radius $R$; the eye globe is strictly spherical in all simulations. Computationally, inducer and responder growth mode differ in the calculation of the radius $R$.

We define the inducer growth mode as growth of the eye globe controlled by the cells in the tissue under consideration. For the computational implementation, we assume that every time a cell divides, the eye surface area increases such that the new total number of cells can achieve its target distance equilibrium without any limitation from the available space. In other words, the eye globe grows just enough to generate the surface area required for housing all cells. Thus, we use the following growth equation in simulations of the inducer growth mode:

$$R(t) = \sqrt{\frac{N_{\text{cells}}(t)(r\delta_{\text{ol}_{\max}})^2}{2}}, \tag{5}$$

where $N_{\text{cells}}(t)$ is the total number of cells at simulation step $t$ that emerges from the simulation. This equation is constructed analogously to **Appendix 1—equation (1)**, and here too we take into consideration the optimal overlap between cells $\delta_{\text{ol}_{\max}}$. Since in our model there is no cell death, $N_{\text{cells}}(t)$ never decreases, and likewise the radius $R$ never decreases.

We define the responder growth mode as growth of the eye globe independent from the cells in the tissue under consideration. Thus, we formulate the growth equation:

$$R(t) = R_{\text{init}} + c_{\text{R}}t, \tag{6}$$

and set $c_{\text{R}}$ as a constant. In short, the radius of the eye globe grows at a constant linear rate in all simulations with the responder growth mode.

## 2.4 Cell division

The flowchart in **Figure 2—figure supplement 1** summarizes the decisions that govern cell proliferation that will be described in the following paragraphs.

Proliferative cells commit to cell division with a probability $p_{\text{division}}$ at every simulation step. If the minimum cell cycle time $t_{\text{cellCycle}}$ has not been attained the division is delayed until this

time elapsed, otherwise cells divide immediately. As in other cell-center agent based models, we introduce a rule that forbids cell division if the local cell density is too high (*Pathmanathan et al., 2009*; *Osborne et al., 2017*; *Sütterlin et al., 2017*). We implement this rule as follows: We calculate local cell packing for a cell $c$ as the average overlap to all neighboring cells $n_i$; cell division is not permitted if the average overlap exceeds a threshold proportional to the cell diameter

$$\frac{\sum_{i=1}^{n} d_{ol}(\mathbf{r}_c, \mathbf{r}_{n_i})}{n} > 2r\delta_{\text{ol\_threshold}}, \tag{7}$$

where $d_{ol}(\mathbf{r}_c, \mathbf{r}_{n_i})$ is the overlap between cell $c$ and neighboring cell $n_i$ that emerges from the simulation, and $\delta_{\text{ol\_threshold}}$ is a model parameter.

Together with the function for the growth of the eye globe (section 2.3), the value for the overlap threshold $\delta_{\text{ol\_threshold}}$ determines the main difference between inducer and responder growth modes. A small value means that a smaller average overlap is sufficient to arrest the cell cycle (cells arrest at lower densities).

For considerations on the parameter values for $\delta_{\text{ol\_threshold}}$ see section 3.2.

## 2.5 Positioning of daughter cells after division

The introduction of new cells into the simulation follows the general procedure used in cell-center agent based models as previously described (Sütterlin *et al.*, 2017). Briefly, when a cell $j$ located at $\mathbf{r}_j = (\begin{matrix} x_j & y_j & z_j \end{matrix})^{\mathbf{T}}$ divides, a new cell $k$ is introduced into the simulation at position $\mathbf{r}_k = (\begin{matrix} x_k & y_k & z_k \end{matrix})^{\mathbf{T}}$. The initial distance between cells $|\mathbf{r}_k - \mathbf{r}_j|$ is chosen to be a small non-zero value. The coordinates $\mathbf{r}_j$ and $\mathbf{r}_k$ are fed as initial input to the biomechanical model, which then calculates how force balance repositions the cells. This means that initially, the two daughter cells almost completely overlap and then gradually separate, displacing any neighboring cells in a "domino effect". Thus, the final position of the daughter cells at the beginning of the simulation step following division may not fully correspond to the initial position that is calculated upon division, but is biased by it.

In this work, we introduce modifications in the calculation of $\mathbf{r}_k$ to allow cells to divide along a given pre-determined direction, which we explain in the following.

By default, cells divide with random division axis, where we calculate $\mathbf{r}_k$ as

$$\mathbf{r}_k = \begin{pmatrix} x_j \\ y_j \\ z_j \end{pmatrix} + \rho \begin{pmatrix} X - 0.5 \\ X - 0.5 \\ X - 0.5 \end{pmatrix}, \tag{8}$$

where $X$ is a uniformly distributed random number in the interval $[0, 1]$, and $\rho$ is a scaling constant that defines the maximum initial distance between daughter cells.

In simulations with directed division axes, we use different calculations for radial and circumferential divisions. When cell $j$ divides radially, we calculate $\mathbf{r}_k$ as

$$\mathbf{r}_k = \begin{pmatrix} x_j \\ y_j \\ z_j \end{pmatrix} + Y\frac{\rho}{2} \begin{pmatrix} 1 \\ 0 \\ 0 \end{pmatrix} \tag{9}$$

where $Y$ is a number chosen uniformly at random from the set $(-1, 1)$. When cell $j$ divides circumferentially, we calculate $\mathbf{r}_k$ as

$$\mathbf{r}_k = \begin{pmatrix} x_j \\ y_j \\ z_j \end{pmatrix} + Y\frac{\rho}{2} \begin{pmatrix} 0 \\ 1 \\ 1 \end{pmatrix}. \tag{10}$$

The probability for a cell $j$ to choose a radial or circumferential division axis depends on probabilities based on geometry derived in the following.

A hemispherical eye of radius $R$ has an area of

$$A_{\text{eye}} = 2\pi R^2. \tag{11}$$

The CMZ forms a band of width $w$ at the base of the eye, and has an area of

$$A_{\text{CMZ}} = 2\pi R w. \tag{12}$$

Thus, the area ratio between the eye without the CMZ and the CMZ is

$$\frac{A_{\text{eye}} - A_{\text{CMZ}}}{A_{\text{CMZ}}} = \frac{R - w}{w} = \frac{R}{w} - 1. \tag{13}$$

To obtain $\frac{R}{w} - 1$ radial divisions for every circumferential division, we formulate the probability of a radial division as

$$p_{\text{rad}} = \frac{\frac{R}{w} - 1}{\frac{R}{w} - 1 + 1} = 1 - \frac{w}{R}, \tag{14}$$

and the probability for circumferential divisions as

$$p_{\text{circ}} = 1 - p_{\text{rad}} = \frac{w}{R}. \tag{15}$$

## 2.6 Simulations with differential divisions in the ventral sector

We define the ventral sector as a 90° sector on the hemisphere (*Figure 6—figure supplement 2*). To determine if a given cell $c$ located at $\mathbf{r}_c$ lies in this sector, we first calculate the radius of a small circle on the hemisphere enlarged by the radius of the cell $c$:

$$R_{\text{small}}(t) = \sqrt{(R(t) + r)^2 - (\mathbf{r}_{c_1} - \mathbf{s}_1)^2}, \tag{16}$$

where $r$ is the cell radius, $R$ is the radius of the hemisphere, and $\mathbf{r}_{c_1}$ and $\mathbf{s}_1$ denote the x-component of $\mathbf{r}_c$ (cell position) and s (center of eye globe), respectively. A cell lies in the ventral sector if the following holds:

$$\mathbf{r}_{c_2} < \mathbf{s}_2 - \sin(45°) R_{\text{small}}(t), \tag{17}$$

$\mathbf{r}_{c_2}$ and $\mathbf{s}_2$ denote the y-component of $\mathbf{r}_c$ and s, respectively.

In simulations with a lineage-independent extrinsic signal, cells in the non-ventral and ventral sectors are redefined at every simulation step according to *Appendix 1—equations (16-17)*. Cells in the non-ventral sector choose their division axis according to *Appendix 1—equation (9)*, while cells in the ventral sector with a circumferential division axis bias use *Appendix 1—equation (10)* (defaulting to *Appendix 1—equation (9)* in simulations without the circumferential division axis bias).

In simulations with a lineage-bound intrinsic signal, *Appendix 1—equations (16-17)* are used in simulation step 0 to define which lineages belong to the ventral and non-ventral sectors. All progeny of non-ventral lineages choose their division axis according to *Appendix 1—equation (9)*. The division axis of progeny of ventral lineages defaults to *Appendix 1—equation (9)* unless they have a circumferential division axis bias, in which case they follow *Appendix 1—equation (10)*.

## 2.7 Cell differentiation

We implement two cell types in the virtual eye: differentiated cells and proliferative (stem) cells. All divisions produce identical proliferative daughter cells. The fate of cells depends on their position on the virtual eye hemisphere. A cell $c$ at $\mathbf{r}_c$ becomes a differentiated cell type if it moves beyond the width of the CMZ:

$$\mathbf{r}_{c_1} > \mathbf{s}_1 + w, \tag{18}$$

where $\mathbf{r}_{c_1}$ and $\mathbf{s}_1$ denote the x-component of $\mathbf{r}_c$ (cell position) and $\mathbf{s}$ (center of eye globe), respectively. Differentiated cells cannot revert to proliferative cells.

## 3 Model parameters

Parameter values used in the simulations presented in this work are listed in *Appendix 1—table 1*. Unless otherwise stated, we used the same parameter values for all simulations.

**Appendix 1—table 1.** Model parameters. Parameters for the force balance calculation of the biomechanical model are identical to previous work (*Sütterlin et al., 2017*) and are not listed.

| Description | Parameter | Value | Reference/Explanation |
|---|---|---|---|
| Biomechanical model parameters | | | |
| Biomechanical calculation step. | $\Delta t$ | 36s | (*Sütterlin et al., 2017*) |
| Seconds per simulation step. | $t_{simstep}$ | $3600s[simstep]^{-1}$ | (*Sütterlin et al., 2017*) |
| Optimal overlap for obstacle cells. | $\delta_{ol_{obstacleCells}}$ | 0.5 | Determined by parameter scan to create a tight barrier to cell movement. |
| Optimal overlap for retinal cells. | $\delta_{ol_{max}}$ | 0.85 | (*Sütterlin et al., 2017*) |
| Initial distance between daughter cells. | | $0.005\mu m$ | (*Sütterlin et al., 2017*) |
| Initial condition parameters | | | |
| Initial radius of eye globe. | $R_{init}$ | $100\mu m$ | Estimated from preparations of hatchling eyes. |
| Minimal displacement threshold. | $\mu$ | $0.2\mu m$ | Determined by parameter scan to generate even initial cell distribution. |
| Simulation parameters | | | |
| Retinal cell radius. | $r$ | $3.5\mu m$ | Estimated from histological sections. |
| Width of the stem cell domain. | $w$ | $25\mu m$ | Estimated from histological sections. |
| Overlap threshold beyond which cell cycle is arrested. | $\delta_{ol\_threshold}$ | 0.4 | *Value for inducer growth mode.* Estimated from parameter scan to minimize density-dependent cell cycle arrest. |
| | | 0.2 | *Value for responder growth mode.* Estimated from parameter scan to maximize density-dependent cell cycle arrest without completely suppressing division. |
| Minimal cell cycle length. | $t_{cellCycle}$ | 24h | Chosen to produce a plausible biological growth rate. |
| Probability of cell division. | $p_{division}$ | $\frac{1}{26}h^{-1}$ | Chosen to produce a plausible biological growth rate. |
| | | $\frac{1}{52}h^{-1}$ | *Value for ventral lineages with differential behavior.* |
| Growth rate of the eye radius (*only in responder growth mode*). | $c_R$ | $6.94 \cdot 10^{-5}\mu m\ s^{-1}$ | Chosen as a small value to ensure quasi-static growth within the biologically plausible growth rate range. |

DOI: https://doi.org/10.7554/eLife.42646.047

In the following sections, we discuss the rationale for choosing parameter values that are not fixed by experimental observations.

### 3.1 Minimal displacement threshold

To ensure that cells are well-distributed on the hemispherical surface for the initial condition of the simulation, we place a predetermined number of cells on the surface and simulate the biomechanical model until cell displacement minimizes (see section 2.1). Numerical fluctuations lead to a small baseline cell displacement, therefore requiring a threshold cut-off value, which we call $\mu$. In the absence of such a threshold, the simulation converges to a value of $\mu = 0.07$ µm average cell displacement per biomechanical model simulation step (1/100 of a cell's diameter; *Appendix 1—figure 1E*). Values of $\mu$ between $0.7$ µm $- 0.2$ µm result in similar arrangements of evenly-distributed cells (*Appendix 1—figure 1A and B*). At $\mu = 0.7$ µm cells failed to completely cover the hemisphere, leaving a small gap (*Appendix 1—figure 1C*). At $\mu = 20$ µm there is no biomechanical calculation and cells were unequally distributed with local dense foci and large empty spaces (*Appendix 1—figure 1D*). To minimize the calculation time while still obtaining an even cell distribution, we chose $\mu = 0.2$ µm.

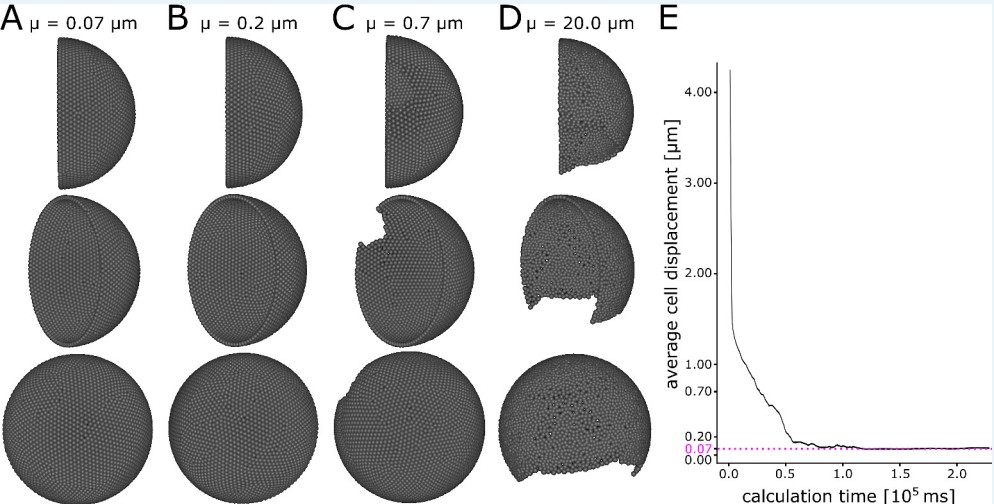

**Appendix 1—figure 1.** A minimum displacement threshold µ = 0.2 ensures even cell distribution. Different views of initial condition of the simulation with (**A**) µ = 0.07 µm, (**B**) µ = 0.2 µm, (**C**) µ = 0.7 µm, (**D**) µ = 20.0 µm. (**E**) Calculation time plotted against the average cell displacement during initialization of the simulation. The simulation converges to 0.07 µm average displacement (pink dashed line).
DOI: https://doi.org/10.7554/eLife.42646.048

### 3.2 Overlap threshold

In the complete absence of coupling between cell division and eye growth, the growth rate may exceed cell production rate, resulting in few cells dispersed over a large surface (*Appendix 1—figure 2 A'-A''''*). In the opposite case where cell production exceeds organ growth rate, cells become packed to a physically implausible degree (*Appendix 1—figure 2 B'-B''''*). As cell density and thus inter-cell forces increase, some cells escape through the layer of obstacle cells and proliferate on the unused half of the sphere (*Appendix 1—figure 2 B''* inset, B'''-B''''*).

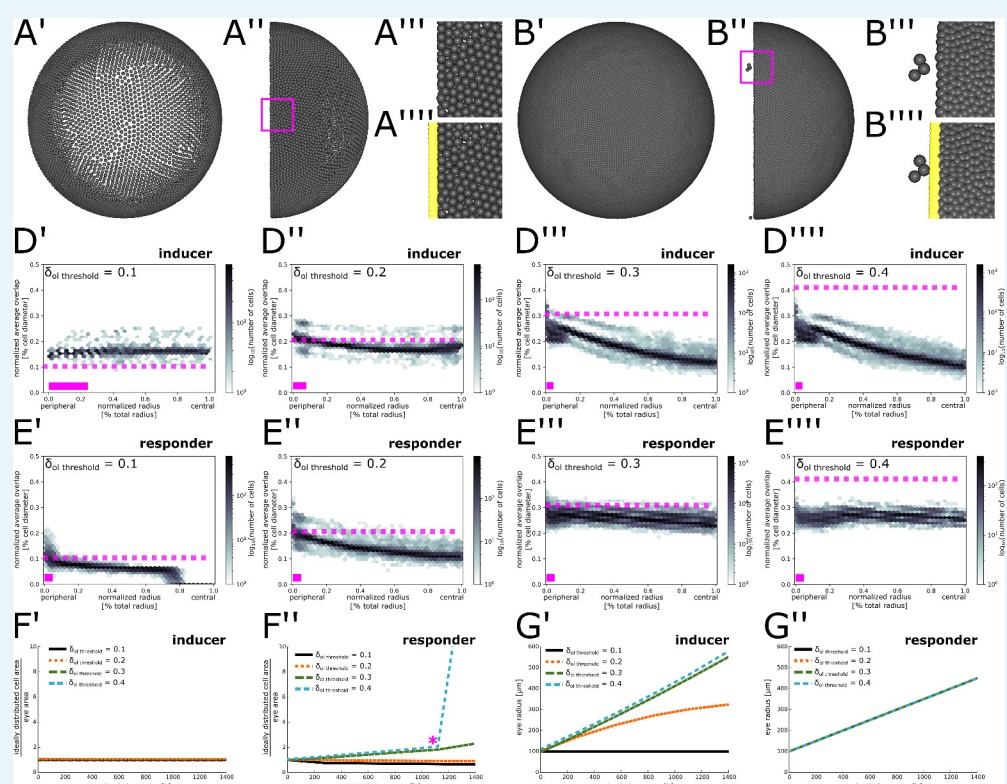

**Appendix 1—figure 2.** Parameter scan to determine optimal overlap threshold values. (**A′–B′′′′**) Different views of representative simulations lacking coupling between eye radius growth and cell proliferation. (**A′–A′′**) Eye area growth rate exceeds cell proliferation rate, resulting in cell dispersion. (**A′′′–A′′′′**) Magnification of inset in (**A′′**) showing peripheral cells without (**A′′′**) or with (**A′′′′**) obstacle cells displayed. (**B′–B′′**) Cell proliferation rate exceeds eye area growth rate, resulting in extremely dense cell packing. (**B′′′–B′′′′**) Magnification of inset in (**B′′**) showing peripheral cells without (**B′′′**) or with (**B′′′′**) obstacle cells displayed. Three cells have squeezed through the obstacle cell layer. (**D′–E′′′′**) Normalized average overlap of cells at simulation step 1400 plotted against their position along the normalized radius. Dashed pink line: Value of $\delta_{\mathrm{ol\_threshold}}$ used for the respective simulation. Solid pink bar: Extent of virtual CMZ. (**D–D′′′′**) inducer growth mode; (**E′–E′′′′**) responder growth mode. (**F′–F′′**) Ratio between total area required by cells and total eye area from simulation step 0 to simulation step 1400 for different values of $\delta_{\mathrm{ol\_threshold}}$. (**F′**) inducer growth mode; (**F′′**) responder growth mode. Pink asterisk marks approximate time when cells start squeezing through obstacle cell layer. (**G′–G′′**) Growth of the eye radius from simulation step 0 to simulation step 1400 for different values of $\delta_{\mathrm{ol\_threshold}}$. (**G′**) inducer growth mode; (**G′′**) responder growth mode.

DOI: https://doi.org/10.7554/eLife.42646.049

To couple eye growth to cell proliferation, we introduce an overlap threshold $\delta_{\mathrm{ol\_threshold}}$ (see sections 2.3 and 2.4). Cells in the inducer growth mode are, by definition, less sensitive to density-dependent arrest, while cells in the responder growth mode are more sensitive. This means that the overlap threshold in the inducer growth mode should exceed the overlap threshold in the responder growth mode: $\delta_{\mathrm{ol\_threshold\_inducer}} > \delta_{\mathrm{ol\_threshold\_responder}}$. We performed a parameter scan (*Appendix 1—figure 2 D′-G′′*) to determine values for $\delta_{\mathrm{ol\_threshold}}$ such that:

i. In the inducer growth mode, density-dependent cell cycle arrest is minimal.
ii. In the responder growth mode, density-dependent cell cycle arrest is maximal **and** cells are not completely arrested.

In the inducer growth mode, growth of the radius depends on the total number of cells in the simulation (*Appendix 1—equation (5)*). Thus, growth rate increases as the overlap threshold is increased (*Appendix 1—figure 2 G'*). A value of $\delta_{\mathrm{ol\_threshold}} = 0.1$ completely inhibits cell proliferation, as the equilibrium average overlap normalized to the cell diameter (normalized average overlap) exceeds the threshold value (*Appendix 1—figure 2 D'*). As a result, the organ does not grow at all (*Appendix 1—figure 2 G'*, solid black line). At $\delta_{\mathrm{ol\_threshold}} = 0.2$, a large population of cells in the proliferative domain (located at the periphery of the radius; indicated by solid pink bar) exceeds the threshold (*Appendix 1—figure 2D''*). Generation of new cells through division increases the local cell density, resulting in inhibition of proliferation due to the low overlap threshold and a gradual reduction in growth rate (*Appendix 1—figure 2 G'* orange dashed line). At $\delta_{\mathrm{ol\_threshold}} = 0.3$, only few cells exceed the threshold (*Appendix 1—figure 2 D'''*), and growth is almost unconstrained (*Appendix 1—figure 2 G'* green dashed line). At $\delta_{\mathrm{ol\_threshold}} = 0.4$, no cells exceed the threshold and growth is completely unconstrained (*Appendix 1—figure 2 G'* cyan dashed line). Thus, this condition fulfills requirement I. At all values of $\delta_{\mathrm{ol\_threshold}}$, the ratio between total area required by cells and the hemisphere area (area ratio) is equal to one throughout the simulation, meaning that - on average - cells are evenly distributed and ideally packed (*Appendix 1—figure 2 F'*).

In the responder growth mode, the radius of the hemisphere steadily grows regardless of the number of cells in the simulation (*Appendix 1—figure 2 G''*). A value of $\delta_{\mathrm{ol\_threshold}} = 0.1$ strongly inhibits cell proliferation, but as the radius grows cells become dispersed and eventually go under the threshold allowing some proliferation (*Appendix 1—figure 2 E'*). However, the area ratio of cells to hemisphere steadily decreases indicating the formation of inter-cell gaps (*Appendix 1—figure 2 F''* solid black line). At $\delta_{\mathrm{ol\_threshold}} = 0.2$, many, but not all, cells are inhibited (*Appendix 1—figure 2 E''*), and the area ratio is near one throughout the simulation (*Appendix 1—figure 2 F''* orange dashed line). At $\delta_{\mathrm{ol\_threshold}} = 0.3$ and $\delta_{\mathrm{ol\_threshold}} = 0.4$, cell proliferation overtakes area growth, resulting in high cell packing all over the hemisphere (*Appendix 1—figure 2 E''' and E''''*). As a result, the area ratio increases over time (*Appendix 1—figure 2 F''* green and cyan dashed lines), until cell packing becomes so severe that cells escape through the obstacle cell layer and proliferate exponentially on the unused half of the sphere (*Appendix 1—figure 2 F''* pink asterisk). Given these data, a value of $\delta_{\mathrm{ol\_threshold}} = 0.2$ best fulfills requirement II while generating an even distribution of cells on the hemisphere for the full duration of the simulation.

## 3.3 Proliferation probability, minimum cell cycle, and growth rate of the retinal radius

A hatchling medaka grows to sexual maturity within 2–3 months. Growth rates vary between individuals, and retinae recovered from young adult fish have radii in the range of 600–800 µm. During this period, if fish are regularly fed and reared at low individual density, growth is approximately linear (*Appendix 1—figure 3 A*), so the growth rate of the retinal radius can be estimated to lie in the following range:

$$\frac{600[\mu\mathrm{m}] - 100[\mu\mathrm{m}]}{90 \cdot 24[\mathrm{h}]} \approx 0.23 \frac{\mu\mathrm{m}}{\mathrm{h}} \tag{19}$$

$$\frac{800[\mu\mathrm{m}] - 100[\mu\mathrm{m}]}{60 \cdot 24[\mathrm{h}]} \approx 0.49 \frac{\mu\mathrm{m}}{\mathrm{h}}. \tag{20}$$

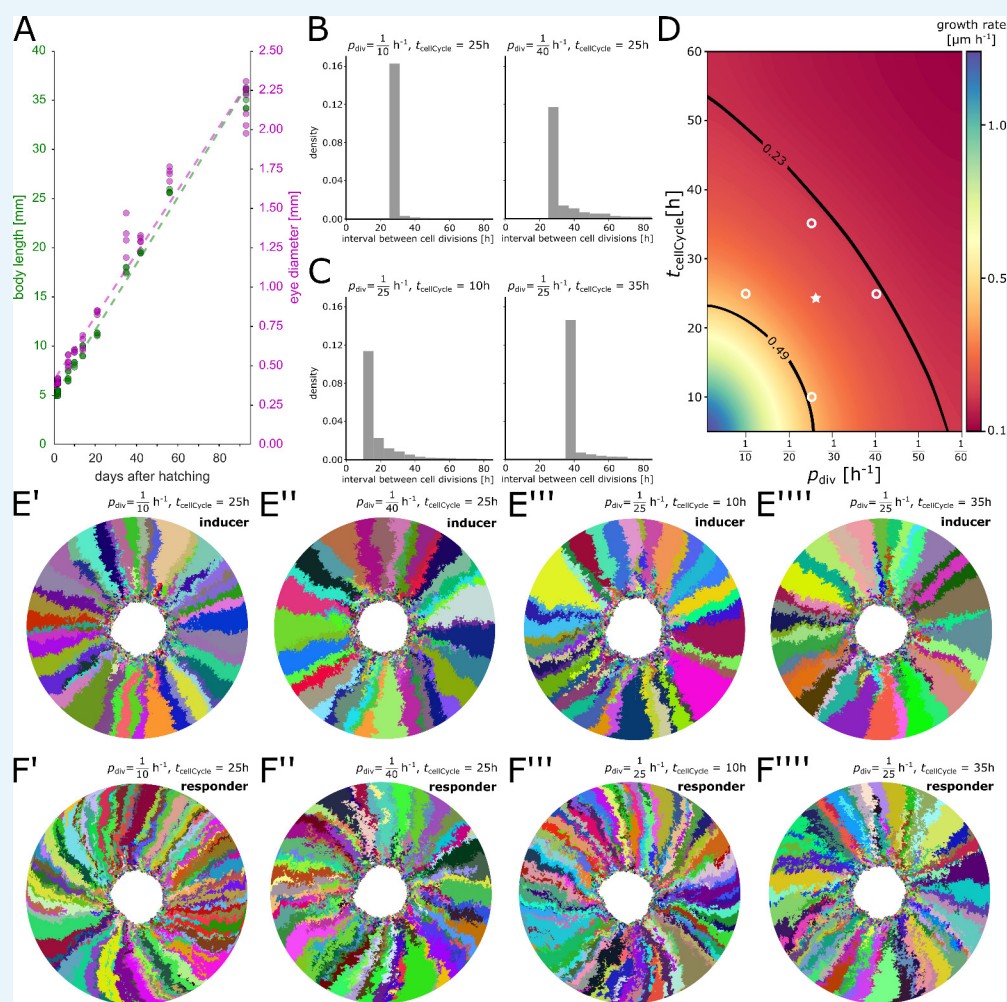

**Appendix 1—figure 3.** Parameter scan of minimum cell cycle and division probability. (**A**) Experimental data. Both body length (green) and eye diameter (magenta) grow approximately linearly over the first 90 days after hatching. (**B**) Distribution of cell division intervals with fixed $t_{cellCycle}$ and variable $p_{division}$. (**C**) Distribution of cell division intervals with variable $t_{cellCycle}$ and fixed $p_{division}$. (**D**) Eye growth rates in the simulation determined from a parameter scan of $t_{cellCycle}$ and $p_{division}$ entailing over 150 simulation runs; intermediate values were interpolated. The plausible parameter space estimated from experimental measurements is contoured by black lines. Open white circles represent values for simulations depicted in (**B, C, E′–E′′′′, F′–F′′′′**). White star represents values used for simulations in the main manuscript. (**E′–E′′′′**) Representative simulations of the inducer growth mode at different values for $t_{cellCycle}$ and $p_{division}$. (**F′–F′′′′**) Representative simulations of the responder growth mode at different values for $t_{cellCycle}$ and $p_{division}$. Throughout the figure $p_{division}$ is abbreviated as $p_{div}$.

DOI: https://doi.org/10.7554/eLife.42646.050

Cell division intervals are not characterized in post-embryonic retinal stem cells. In other proliferative stem cell systems, such as mouse tail skin, cell division intervals follow a right-skewed distribution, which can be modelled by combining a minimum division interval with a fixed probability for division (**Klein et al., 2007**). In the absence of a minimum cell cycle, division intervals in this model follow an exponential distribution, and would thus allow manifestly unphysiological cell cycle times that can be arbitrarily short (**Klein et al., 2007**).

Our cell proliferation model assumes that a cell may commit to cell division at any time with probability $p_{division}$, but must wait a minimum of $t_{cellCycle}$ simulation steps before actually

dividing. These rules generate a distribution of cell cycle intervals with a peak at $t_{cellCycle}$ and exponential decrease thereafter (*Appendix 1—figure 3 B-C*). The magnitude of the peak and the exponential decay increase with increasing $p_{division}$ (*Appendix 1—figure 3B*). As expected, increasing $t_{cellCycle}$ shifts the distribution to the right, and also increases the peak (*Appendix 1—figure 3C*).

By parameter scan, we determined which combinations of $p_{division}$ and $t_{cellCycle}$ result in growth rates within the range in *Appendix 1—equations (19-20)* (*Appendix 1—figure 3D*). Different values for $p_{division}$ and $t_{cellCycle}$ in inducer and responder growth mode resulted in qualitatively similar clone properties (*Appendix 1—figure 3E'-E''''* (inducer), F'-F'''' (responder)). The parameters we chose for the simulations presented in the manuscript fall in the middle of this biologically plausible range (white star in *Appendix 1—figure 3D*).

