## [Decision Letter]

[Editors’ note: the authors were asked to provide a plan for revisions before the editors issued a final decision. What follows is the editors’ letter requesting such plan.]

Thank you for sending your article entitled "Retinal stem cells modulate proliferative parameters to coordinate post-embryonic morphogenesis in the eye of fish" for peer review at *eLife*. Your article is being evaluated by Marianne Bronner as the Senior Editor, a Reviewing Editor, and two reviewers.

Given the list of essential revisions, including new experiments, the editors and reviewers invite you to respond within the next two weeks with an action plan for the completion of the additional work. We plan to share your responses with the reviewers and then issue a binding recommendation.

Summary:

In this paper, Tsingos et al. use imaging and modelling approaches to understand how an organ (the eye) can continually grow throughout an organism's life while maintaining a precise morphology. They identify a novel interplay between neural retina and retinal pigmented epithelium cells, whereby the latter divide more stochastically. They further identify orientated cell divisions as critical in regulating the organ growth. They effectively utilise modelling to demonstrate how these different modes of cell differentiation and division can result in coordinated continual organ growth.

Essential revisions:

1) A first major comment is on fundamental feedback modes. The authors state that the feedback mechanisms coordinating growth of all tissues could be wired in "two ways: either the tissue of interest acts upstream to induce growth of other tissues, or, vice versa, the tissue of interest lies downstream of growth cues from another tissue in the organ", named respectively "inducer growth mode" and "responder growth mode". It is unclear upon what evidence this dichotomy bears. Specifically, in comparison to these extreme cases, in how far would the model be able to distinguish between more intermediate scenarios, where growth is regulated both by extrinsic and intrinsic cues, or both tissues serve as inducer and responder at the same time?

2) The impact of this study is limited by a lack of statistical and biophysical rigor. The authors make experimental observations that are, in principle, quantitative. However, the predictions made by the computational model are only qualitatively compared to experimental data. In the revised version of the manuscript the authors should (a) employ statistical tests and calculate p-values to quantify the statistical significance of biological effects, (b) quantitatively compare experimental observations (e.g. clone fragment size distributions or similar) with computational predictions, and (c) employ appropriate statistical criteria to evaluate whether the model is quantitatively able to distinguish between different biological scenarios.

3) On the biophysical side, the computational model is based on a rather complex set of rules which depend on a large number of parameters. Many of these parameters are fixed based on experimental measurements, but some, in particular those changed between the two biological scenarios, do not have a clear biophysical interpretation or are adjusted "empirically". For example, the probability of division for virtual stem cells is set to be 1/26 per hour. It is unclear how this value is chosen. It is also inadequately explained how the minimal displacement threshold, µ, is "empirically determined". It has to be shown that the main conclusions of this work do not sensitively depend on the specific values of parameters that are not experimentally measured.

4) The concept of stochasticity in proliferation is used in a confusing way throughout the text. In subsection “Fundamental feedback modes of organ and cell growth impact on clonal patterns”, the authors imply that the responder growth mode leads to higher variability of cell cycle timing in individual cells. As cell divisions in both growth modes are, by the definition of the model, stochastic the authors should use more appropriate statistical wording for this finding (e.g. variance of cell cycle times). In addition, this result should be quantitatively demonstrated based on simulation data. Further, the authors do not comment on the biophysical origins of the observed differences in the variability of cell cycle timing. Are they associated with fluctuations in cell density?

5) (Statistics) The results of Figure 1E are promising but the analysis can be strengthened. Presentation of a single image is not sufficient to convince the reader that RPE ArCoS frequently have more irregular shaped clones. Quantification of the clonal topology across a number of samples is necessary to provide meaningful support. A similar critique holds for Figure 2D' – superficially, the clusters do not look that different and better quantification is needed to backup such statements (subsection “Fundamental feedback modes of organ and cell growth impact on clonal patterns”).

6) The description of the model can be improved. Details are too vague. For example, I found the sentence "We took advantage of this…" (subsection “A minimal complexity 3D agent based model of retinal tissues”) unclear – what exactly is helping and why? It is not obvious to me how you go from discussing the multi-layered columns of the progeny to simplifying model complexity. Further, clearer justification needs to be given why an effectively hard sphere model is a reasonable approximation to an epithelial tissue.

7) The authors argue that δ = 0.4 is the right threshold for overlap. It would be helpful to more explicitly elucidate what happens to the system as δ is increased or decreased – this would provide a clearer picture of how the parameter δ regulates the model output.

8) The nomenclature for the ratios is incorrect. Typically, a colon ": " is used to represent a ratio, not an "=" sign. This makes the results in subsection “NR stem cells undergo radial divisions at the rate predicted by shape regulation” hard to read and this needs to be reworked to improve clarity.

9) Subsection “NR stem cells undergo radial divisions at the rate predicted by shape regulation – the ArCoS width (which implies a distance) is given as an angle. It is either the angular extent (or similar) or give the arc length. Relatedly, I cannot believe that the experiments are measured to a hundredth of a degree (4.87º). Give experimental measures (here and elsewhere) to an appropriate level of accuracy (in this case, likely 5º).

10) Subsection “NR stem cells undergo radial divisions at the rate predicted by shape regulation”. The authors note that simulations with "ideal" division axis match the experiments. What happens at more intermediate levels – i.e. how robust is the system?

11) Figure 6E (subsection “Local biases in ventral NR stem cell divisions influence retinal topology”) does not seem to be present.

---

## [Author Response]

[Editors' note: the authors’ plan for revisions was approved and the authors made a formal revised submission.]

Essential revisions:1) A first major comment is on fundamental feedback modes. The authors state that the feedback mechanisms coordinating growth of all tissues could be wired in "two ways: either the tissue of interest acts upstream to induce growth of other tissues, or, vice versa, the tissue of interest lies downstream of growth cues from another tissue in the organ", named respectively "inducer growth mode" and "responder growth mode". It is unclear upon what evidence this dichotomy bears. Specifically, in comparison to these extreme cases, in how far would the model be able to distinguish between more intermediate scenarios, where growth is regulated both by extrinsic and intrinsic cues, or both tissues serve as inducer and responder at the same time?

We thank the referees for pointing this out. It seems we failed to properly contextualize the inducer/responder concept in the submitted manuscript. We have written a new opening paragraph in the Introduction which reviews relevant literature on growth control, and reworded the passage in the Results section to clarify our concept of tissue growth modes and possible biological mechanisms.

As starting point of a dynamically evolving model we had deliberately selected two reference scenarios to be sure to cover the entire spectrum. This allowed to anchor the experimentally determined behavior of neural retinal (NR, inducer) as well as retinal pigmented epithelial (RPE, responder) stem cells. Our model does not require a predefinition of regulatory interactions and consequently would detect intermediate scenarios. Our and others’ (Zhang et al., 2016; cited in Discussion section) experimental data are evidence for inducer and responder growth modes in tissues growing co-ordinately. Since the model aimed at addressing the mode of growth rather than the nature of the contributing cues, the implementation of aspects such as contribution of extrinsic and intrinsic cues (even though interesting) is beyond the scope of this work.

In the revised version of the manuscript we have taken care to improve the precision of wording of the passage to explain the rationale behind our modelling choices and contextualize them better.

2) The impact of this study is limited by a lack of statistical and biophysical rigor. The authors make experimental observations that are, in principle, quantitative. However, the predictions made by the computational model are only qualitatively compared to experimental data. In the revised version of the manuscript the authors should (a) employ statistical tests and calculate p-values to quantify the statistical significance of biological effects, (b) quantitatively compare experimental observations (e.g. clone fragment size distributions or similar) with computational predictions, and (c) employ appropriate statistical criteria to evaluate whether the model is quantitatively able to distinguish between different biological scenarios.

The referees are right, all the quantitative data for statistical analysis had already been determined in the initially submitted version without presenting them prominently. We are happy to amend the figures to include the result of statistical tests in the respective figures. Already in the initially submitted version we had selected representative data points for illustration combined with statistical analyses. Following the referees’ advice, we have now employed statistical tests to compare experiment and simulation and provided the respective p-values in the revised version of the manuscript as detailed below.

a) As the referees requested, we have included the result of statistical tests for the quantifications in Figure 3, Figure 4, Figure 5 and Figure 6. For Figure 3, the p-values are listed for all conditions in Figure 3—source data 1, Figure 3—source data 2 and Figure 3—source data 3. We included a new quantification in Figure 4 to directly compare experiment, and have included the p-values in the new panel F. For Figure 5, the p-values were added as an inset in panel D’. For Figure 6, p-values are given in Figure 6—figure supplement 1D’-D’’’’ along with a density plot of the quantified data. We have adapted the text where appropriate and reworded sentences to reflect the result of the calculation, which support and strengthen the main findings of our manuscript.

b) Quantification of several clone fragment properties in experimental and simulated data were included in Figure 3, Figure 4, and Figure 5 of the submitted version of the manuscript. We have reworded the text to better reflect this. As the referees suggested, we also included plots of clone fragment distribution in Figure 3—figure supplement 2, and Figure 6—figure supplement 1. Additionally, we also included new quantification data for simulations in Figure 2 (see also response to essential revision #4), and between experiment and simulation in Figure 4.

c) See point (a).

3) On the biophysical side, the computational model is based on a rather complex set of rules which depend on a large number of parameters. Many of these parameters are fixed based on experimental measurements, but some, in particular those changed between the two biological scenarios, do not have a clear biophysical interpretation or are adjusted "empirically". For example, the probability of division for virtual stem cells is set to be 1/26 per hour. It is unclear how this value is chosen. It is also inadequately explained how the minimal displacement threshold, µ, is "empirically determined". It has to be shown that the main conclusions of this work do not sensitively depend on the specific values of parameters that are not experimentally measured.

One of the advantages of the chosen computational model is a rather simple set of rules. The relevant parameters had been determined experimentally by scanning the parameter space by simulations. We thank the referees for pointing out the ambiguity of the term “empirically” and have substituted all occurrences to clarify whether the values were determined experimentally or by repeated simulations.

To better present the rationale for the choice of parameters, we greatly expanded the final section of the appendix (subsection “3. Model parameters”, and three additional appendix figures). Here we have now detailed the rationale for the choice of value of every non-fixed parameter, and provided additional simulations to illustrate the process of parameter choice and the impact of their variation:

– The rationale for choosing the minimal displacement threshold µ has been expanded and illustrated in section 3.1 and in Appendix—figure 1.

– The determination of the appropriate values for the parameter δ_ol_threshold_, are elucidated in section 3.2 and in Appendix—figure 2.

– Supporting experimental data as well as a detailed parameter scan of probability of division and minimum cell cycle are explained in Appendix 1, section 3.3 and Appendix—figure 3.

These critical considerations are referred to and highlighted in the main text of the revised version of the manuscript.

4) The concept of stochasticity in proliferation is used in a confusing way throughout the text. In subsection “Fundamental feedback modes of organ and cell growth impact on clonal patterns”, the authors imply that the responder growth mode leads to higher variability of cell cycle timing in individual cells. As cell divisions in both growth modes are, by the definition of the model, stochastic the authors should use more appropriate statistical wording for this finding (e.g. variance of cell cycle times). In addition, this result should be quantitatively demonstrated based on simulation data. Further, the authors do not comment on the biophysical origins of the observed differences in the variability of cell cycle timing. Are they associated with fluctuations in cell density?

We thank the referees for this helpful comment and advice. The term “stochastic” has different meanings in different fields, and we realize that we used the term ambiguously. We have revised the manuscript to make sure that all ambiguity is removed.

In addition, as the referees suggested, we have generated a heat map showing the distribution of interval duration between two subsequent cell division events in relation to each cell’s mean average overlap during that interval. These data were included for both inducer and responder growth modes in new panels C’’’’ and D’’’’ in Figure 2.

In accordance with our previous qualitative description, these data quantitatively show both that intervals between cell divisions tend to be longer in the responder growth mode, and that cells experiencing larger overlap (i.e. higher cell density) tend to have longer division intervals. These points are now highlighted in the manuscript.

5) (Statistics) The results of Figure 1E are promising but the analysis can be strengthened. Presentation of a single image is not sufficient to convince the reader that RPE ArCoS frequently have more irregular shaped clones. Quantification of the clonal topology across a number of samples is necessary to provide meaningful support. A similar critique holds for Figure 2D' – superficially, the clusters do not look that different and better quantification is needed to backup such statements (subsection “Fundamental feedback modes of organ and cell growth impact on clonal patterns”).

Apparently, we failed to well guide the referees/reader through the initially submitted manuscript. We had considered Figure 1 and Figure 2 as representative examples, while the quantification had been presented in Figure 3.

To make sure that this becomes apparent in the revised version of the manuscript, we have reworded the relevant passages in the Results section to indicate the qualitative character of these observations which we eventually quantify in detail in Figure 3.

We have provided additional quantifications of simulation data to emphasize the difference between the inducer and responder growth modes (Figure 2, new panels C’’’’ and D’’’’). See also response to essential revision #4.

6) The description of the model can be improved. Details are too vague. For example, I found the sentence "We took advantage of this…" (subsection “A minimal complexity 3D agent based model of retinal tissues”) unclear – what exactly is helping and why? It is not obvious to me how you go from discussing the multi-layered columns of the progeny to simplifying model complexity. Further, clearer justification needs to be given why an effectively hard sphere model is a reasonable approximation to an epithelial tissue.

We have taken care to address this crucial point raised by the referees and have made an effort to precisely describe the model and its advantages in the given context.

Regarding subsection “Clonal analysis indicates NR and RPE follow different post-embryonic growth modes”: Due to the fact that a single stem cell gives rise to a clonal string of descendant differentiated cells (ArCoS) that forms an extended column containing all retinal stem cells we have simplified the model to represent a single cell type as proxy for the entire column at a given position. We have rephrased the passage (now subsection “A 3D agent based model of retinal tissues”) and included a cartoon drawing in Figure 1—figure supplement 2B to better explain our abstraction of the system. We apologize for any confusion caused by our previous wording.

Our choice of using an overlapping spheres model (which, unlike a “hard sphere model” takes deformability of cells into account) is based on the following considerations: Retinal cells form a tightly packed pseudocrystalline mosaic, with very limited cell movement. The bulk of our experimental data consists of clonal labelling, which generate a distribution of outcomes that cannot be described easily with mean-field approaches. The spatial arrangement of cells on a curved surface poses a further challenge. These points can be addressed using agent based approaches. Among agent based approaches, a centre-based overlapping spheres model is the most economical in terms of number of parameters, model assumptions, and calculation time:

– Cellular automata cannot represent curved surfaces well and is arguably too coarse-grained.

– Cellular Potts would require an extremely fine mesh to approximate a curved surface, leading to prohibitively long calculation times, and additional parametrization for cellular growth rates and motility (which is intrinsic to Potts models).

– Vertex-based models are seldom used in 3 dimensions due to the large number of points that need to be calculated per cell, and would require additional parametrization for vertex transitions, intracellular pressure and line tensions along a cells’ edges.

Finally, we note that both stratified and simple epithelia – in particular involving stem cell dynamics and clonal data – have been previously successfully modelled by many groups using a cell-centred overlapping spheres model (Osborne et al., 2010, Buske et al., 2011, Li et al., 2013; referenced in subsection “A 3D agent based model of retinal tissues” of the manuscript).

We have taken care to sufficiently clarify these points in the revised version, and have reworked the introductory paragraph to the subsection “A 3D agent based model of retinal tissues “to more explicitly contextualize our modelling.

7) The authors argue that δ = 0.4 is the right threshold for overlap. It would be helpful to more explicitly elucidate what happens to the system as δ is increased or decreased – this would provide a clearer picture of how the parameter δ regulates the model output.

We thank the referees for this suggestion. We greatly expanded the final section of the appendix to include a detailed rationale for the choice of value of each parameter (see also essential revision 3). The effect of varying the parameter δ_ol_threshold_, are elucidated in Appendix 1, section 3.2 and in Appendix—figure 2 (see also the response to essential point #3).

8) The nomenclature for the ratios is incorrect. Typically, a colon ": " is used to represent a ratio, not an "=" sign. This makes the results in subsection “NR stem cells undergo radial divisions at the rate predicted by shape regulation” hard to read and this needs to be reworked to improve clarity.

We thank the referees for pointing out this mistake and have amended the text to adhere to standard notation.

9) Subsection “NR stem cells undergo radial divisions at the rate predicted by shape regulation – the ArCoS width (which implies a distance) is given as an angle. It is either the angular extent (or similar) or give the arc length. Relatedly, I cannot believe that the experiments are measured to a hundredth of a degree (4.87º). Give experimental measures (here and elsewhere) to an appropriate level of accuracy (in this case, likely 5º).

The smallest unit that can be measured in the data is a single pixel. The experimental data used for quantification in Figure 5D’ were acquired at a resolution of 1.6678 µm/pixel (value obtained from the image metadata saved by the microscope). We feel that rounding to two decimal places is appropriate.

Specifically, in Figure 5D’, we quantified the ratio of the width in pixels belonging to a clone divided by the total number of pixels in the circumference (both measurements made on the transformed data by an automated custom ImageJ plugin). Effectively, the metric we measured in Figure 5D’ corresponds to the central angle enclosed by rays travelling from the center of the embryonic retina to the clone borders at every radial position. Schematically, this is illustrated by Figure 5D’’, directly underneath the graph in Figure 5D’.

Obviously, we were unclear in the submitted version of the manuscript, and have now included a more detailed illustration of the coordinate transformation earlier in the main text (Figure 3—figure supplement 2C; see also response to minor point #17), and more details in the Materials and methods section.

As the reviewer suggested, we considered different nomenclature. However, “arc length” has a different definition, and both “angular distance” and “angular extend” have other meanings in astronomy and optics and could cause confusion, especially since our study concerns the eye. Thus, we reasoned that “width” was a more intuitive word particularly in combination with the illustration in Figure 5D’’, and the additional illustration we now provided in Figure 3—figure supplement 2C.

10) Subsection “NR stem cells undergo radial divisions at the rate predicted by shape regulation”. The authors note that simulations with "ideal" division axis match the experiments. What happens at more intermediate levels – i.e. how robust is the system?

Our model constrains the shape of the organ (a deformable organ would require extensive changes to the model which go beyond the scope of this work). For this reason, cell redistribution can cancel out the effect of circumferential divisions (as schematically illustrated in Figure 5A).

Following the reviewer’s suggestion, we show that the model can robustly produce measurable differences in clone width for circumferential division probabilities ranging from 0% to 50% (as shown in the new Figure 5—figure supplement 1). A random division axis corresponds to a circumferential division probability of roughly one third while the ideal division axis corresponds to a diminishing circumferential division probability over time with a maximum of 25% at the very start of the simulation.

11) Figure 6E (subsection “Local biases in ventral NR stem cell divisions influence retinal topology”) does not seem to be present.

The referees are right, in the internal review of the figure, the corresponding labels (lowest panel of Figure 6) got lost. We have amended this mistake and improved text and figure design for readability.